# Controlling Continuous Relaxation for Combinatorial Optimization

**Yuma Ichikawa**
Fujitsu Limited, Kanagawa, Japan
Department of Basic Science, University of Tokyo

## Abstract

Unsupervised learning (UL)-based solvers for combinatorial optimization (CO) train a neural network that generates a soft solution by directly optimizing the CO objective using a continuous relaxation strategy. These solvers offer several advantages over traditional methods and other learning-based methods, particularly for large-scale CO problems. However, UL-based solvers face two practical issues: **(I)** an optimization issue, where UL-based solvers are easily trapped at local optima, and **(II)** a rounding issue, where UL-based solvers require artificial post-learning rounding from the continuous space back to the original discrete space, undermining the robustness of the results. This study proposes a **C**ontinuous **R**elaxation **A**nnealing[1] (**CRA**) strategy, an effective rounding-free learning method for UL-based solvers. CRA introduces a penalty term that dynamically shifts from prioritizing continuous solutions, effectively smoothing the non-convexity of the objective function, to enforcing discreteness, eliminating artificial rounding. Experimental results demonstrate that CRA significantly enhances the performance of UL-based solvers, outperforming existing UL-based solvers and greedy algorithms in complex CO problems. Additionally, CRA effectively eliminates artificial rounding and accelerates the learning process.

## 1 Introduction

The objective of combinatorial optimization (CO) problems is to find the optimal solution from a discrete space, and these problems are fundamental in many real-world applications [Papadimitriou and Steiglitz, 1998]. Most CO problems are NP-hard or NP-complete; making it challenging to solve large-scale problems within feasible computational time. Traditional methods frequently depend on heuristics to find approximate solutions, but they require considerable insights into the specific problems. Alternatively, CO problems can be formulated as integer linear programming (ILP) and solved using ILP solvers. However, ILP solvers lacks scalability for large-scaled problems.

Recently, several studies have used machine learning methods to handle CO problems by learning heuristics. Most of these studies focus on supervised learning (SL)-based solvers [Hudson et al., 2021, Joshi et al., 2019, Gasse et al., 2019, Selsam et al., 2018, Khalil et al., 2016], which require optimal solutions to CO problems as supervision during training. However, obtaining optimal solutions is challenging in practice, and SL-based solvers often fail to generalize well [Yehuda et al., 2020]. Reinforcement learning (RL)-based solvers [Yao et al., 2019, Chen and Tian, 2019, Yolcu and Póczos, 2019, Nazari et al., 2018, Khalil et al., 2017, Bello et al., 2016] avoid the need for optimal solutions but often suffer from notoriously unstable training due to poor gradient estimation and hard explorations [Mnih et al., 2015, Tang et al., 2017, Espeholt et al., 2018]. Unsupervised learning (UL)-based solvers [Schuetz et al., 2022a, Karalias and Loukas, 2020, Amizadeh et al., 2018] have

---

[1]The code is available at https://github.com/Yuma-Ichikawa/CRA4CO.

recently attracted much attention. UL-based solvers follow a continuous relaxation approach, training a UL model to output a *soft solution* to the relaxed CO problem by directly optimizing a differentiable objective function, offering significantly stable and fast training even for large-scale CO problems. Notably, the physics-inspired GNN (PI-GNN) solver [Schuetz et al., 2022a] employs graph neural networks (GNN) to automatically learn instance-specific heuristics and performs on par with or outperforms existing solvers for CO problems with millions of variables without optimal solutions.

While these offer some advantages over traditional and other machine learning-based solvers, they face two practical issues. The first issue is an optimization issues where UL-based solvers are easily trapped at local optima. Due to this issue, Angelini and Ricci-Tersenghi [2023] demonstrated that the PI-GNN solver [Schuetz et al., 2022a] could not achieve results comparable to those of the degree-based greedy algorithm (DGA) [Angelini and Ricci-Tersenghi, 2019] on maximum independent set (MIS) problems in random regular graphs (RRG). Wang and Li [2023] also pointed out the importance of using dataset or history, and initializing the GNN with outputs from greedy solvers to help the PI-GNN solver overcome optimization challenges. This issue is a crucial bottleneck to the applicability of this method across various real-world applications. The second issue relates to the inherent ambiguity of the continuous relaxation approach. This approach necessitates artificial rounding from the soft solution, which may include continuous values, back to the original discrete solution, potentially undermining the robustness of the results. While linear relaxation can provide an optimal solution for original discrete problems on bipartite graphs [Hoffman and Kruskal, 2010], it typically leads to solutions with $1/2$ values, which is known to half-integrality [Nemhauser and Trotter Jr, 1974], in which existing rounding methods [Schuetz et al., 2022b, Wang et al., 2022] completely lose their robustness. For NP-hard problems with graph structures, such as the MIS and MaxCut, semidefinite programming (SDP) relaxations have been proposed as effective approximation methods [Lovász, 1979, Goemans and Williamson, 1995]. However, these approaches rely on rounding techniques, such as spectral clustering [Von Luxburg, 2007], to transform relaxed solutions into feasible ones, which often fails to obtain optimal solutions.

To address these issues, we propose the **C**ontinuous **R**elaxation **A**nnealing (**CRA**). CRA introduces a penalty term to control the continuity and discreteness of the relaxed variables, with a parameter $\gamma$ to regulate the intensity of this penalty term. When the parameter $\gamma$ is small, the relaxed variable tends to favor continuous solutions, whereas a large $\gamma$ biases them toward discrete values. This penalty term also effectively eliminates local optimum. Moreover, a small $\gamma$ forces the loss function to approach a simple convex function, encouraging active exploration within the continuous space. CRA also includes an annealing process, where $\gamma$ is gradually increased until the relaxed variables approach discrete values, eliminating the artificial rounding from the continuous to the original discrete space after learning. In this study, the solver that applies the CRA to the PI-GNN solver is referred to as the CRA-PI-GNN solver. We also demonstrate the benefits of the CRA through experiments on benchmark CO problems, including MIS, maximum cut (MaxCut), and diverse bipartite matching (DBM) problems across graphs of varying sizes and degrees. The experimental results show that the CRA significantly enhances the performance of the PI-GNN solver, outperforming the original PI-GNN solver, other state-of-the-art learning-based baselines, and greedy algorithms. This improvement is achieved by directly optimizing each instance without any history, e.g., previous optimal solutions and the information of other instances. Additionally, these experiments indicate that the CRA accelerates the learning process of the PI-GNN solver. Notably, these results overcome the limitations pointed out by Angelini and Ricci-Tersenghi [2023], Wang and Li [2023], highlighting the further potential of UL-based solvers.

**Notation.** We use the shorthand expression $[N] = \{1, 2, \ldots, N\}$, where $N \in \mathbb{N}$. $I_N \in \mathbb{R}^{N \times N}$ denotes an $N \times N$ identity matrix, $\mathbf{1}_N$ denotes the vector $(1, \ldots, 1)^\top \in \mathbb{R}^N$, and $\mathbf{0}_N$ denotes the vector $(0, \ldots, 0)^\top \in \mathbb{R}^N$. $G(V, E)$ represents an undirected graph, where $V$ is the set of nodes with cardinality $|V| = N$, and $E \subseteq V \times V$ denotes the set of edges. For a graph $G(V, E)$, $A_{ij}$ denotes the adjacency matrix, where $A_{ij} = 0$ if an edge $(i, j)$ does not exist and $A_{ij} > 0$ if the edge is present.

## 2 Background

**Combinatorial optimization.** The goal of this study is to solve the following CO problem.

$$\min_{\boldsymbol{x} \in \{0,1\}^N} f(\boldsymbol{x}; C) \quad \text{s.t.} \quad \boldsymbol{x} \in \mathcal{X}(C),$$

where $C \in \mathcal{C}$ denotes instance-specific parameters, such as a graph $G = (V, E)$, and $\mathcal{C}$ represents the set of all possible instances. $f : \mathcal{X} \times \mathcal{C} \to \mathbb{R}$ denotes the cost function. Additionally, $\boldsymbol{x} = (x_i)_{1 \leq i \leq N} \in \{0, 1\}^N$ is a binary vector to be optimized, and $\mathcal{X}(C) \subseteq \{0, 1\}^N$ denotes the feasible solution space, typically defined by the following equality and inequality constraints.

$$\mathcal{X}(C) = \{\boldsymbol{x} \in \{0, 1\}^N \mid \forall i \in [I], \ g_i(\boldsymbol{x}; C) \leq 0, \ \forall j \in [J], \ h_j(\boldsymbol{x}; C) = 0\}, \ \ I, J \in \mathbb{N},$$

where, for $i \in [I]$, $g_i : \{0, 1\}^N \times \mathcal{C} \to \mathbb{R}$ denotes the inequality constraint, and for $j \in [J]$, $h_j : \{0, 1\}^N \times \mathcal{C} \to \mathbb{R}$ denotes the equality constraint. Following UL-based solvers [Wang et al., 2022, Schuetz et al., 2022a, Karalias and Loukas, 2020], we reformulate the constrained problem into an equivalent unconstrained form using the penalty method [Smith et al., 1997]:

$$\min_{\boldsymbol{x}} l(\boldsymbol{x}; C, \boldsymbol{\lambda}), \ \ l(\boldsymbol{x}; C, \boldsymbol{\lambda}) \triangleq f(\boldsymbol{x}; C) + \sum_{i=1}^{I+J} \lambda_i v_i(\boldsymbol{x}; C).$$

where, for all $i \in [I + J]$, $v : \{0, 1\}^N \times \mathcal{C} \to \mathbb{R}$ is the penalty term, which increases when the constraints are violated. For example, the penalty term is defined as follows:

$$\forall i \in [I], j \in [J], \ v_i(\boldsymbol{x}; C) = \max(0, g_i(\boldsymbol{x}; C)), \ \ \forall j \in [J], \ v_j(\boldsymbol{x}; C) = (h_j(\boldsymbol{x}; C))^2,$$

and $\boldsymbol{\lambda} = (\lambda_i)_{1 \leq i \leq I+J} \in \mathbb{R}^{I+J}$ denotes the penalty parameters that control the trade-off between constraint satisfaction and cost optimization. Note that, as $\boldsymbol{\lambda}$ increases, the penalty for constraint violations becomes more significant. In the following, we provide an example of this formulation.

**Example: MIS problem.**  The MIS problem is a fundamental NP-hard problem [Karp, 2010], defined as follows. Given an undirected graph $G(V, E)$, an independent set (IS) is a subset of nodes $\mathcal{I} \in V$ where any two nodes are not adjacent. The MIS problem aims to find the largest IS, denoted as $\mathcal{I}^*$. In this study, $\rho$ denotes the IS density, defined as $\rho = |\mathcal{I}|/|V|$. Following Schuetz et al. [2022a], a binary variable $x_i$ is assigned to each node $i \in V$. The MIS problem can be formulated as follows:

$$f(\boldsymbol{x}; G, \lambda) = -\sum_{i \in V} x_i + \lambda \sum_{(i,j) \in E} x_i x_j,$$

where the first term maximizes the number of nodes assigned a value of 1, and the second term penalizes adjacent nodes assigned 1 according to the penalty parameter $\lambda$.

## 2.1  Unsupervised learning based solvers

Learning for CO problems involves training an algorithm $\mathcal{A}_{\boldsymbol{\theta}}(\cdot) : \mathcal{C} \to \{0, 1\}^N$ parameterized by a neural network (NN), where $\boldsymbol{\theta}$ denotes the parameters. For a given instance $C \in \mathcal{C}$, this algorithm generates a valid solution $\hat{\boldsymbol{x}} = \mathcal{A}_\theta(C) \in \mathcal{X}(C)$ and aims to minimize $f(\hat{\boldsymbol{x}}; C)$. Several approaches have been proposed to train $\mathcal{A}_\theta$. This study focuses on UL-based solvers, which do not use a labeled solution $\boldsymbol{x}^* \in \operatorname{argmin}_{\boldsymbol{x} \in \mathcal{X}(C)} f(\boldsymbol{x}; C)$ during training [Wang et al., 2022, Schuetz et al., 2022a, Karalias and Loukas, 2020, Amizadeh et al., 2018]. In the following, we outline the details of the UL-based solvers.

The UL-based solvers employ a continuous relaxation strategy to train NN. This continuous relaxation strategy reformulates a CO problem into a continuous optimization problem by converting discrete variables into continuous ones. A typical example of continuous relaxation is expressed as follows:

$$\min_{\boldsymbol{p}} \hat{l}(\boldsymbol{p}; C, \boldsymbol{\lambda}), \ \ \hat{l}(\boldsymbol{p}; C, \boldsymbol{\lambda}) \triangleq \hat{f}(\boldsymbol{p}; C) + \sum_{i=1}^{m+p} \lambda_i \hat{v}_i(\boldsymbol{p}; C),$$

where $\boldsymbol{p} = (p_i)_{1 \leq i \leq N} \in [0, 1]^N$ represents a set of relaxed continuous variables, where each binary variable $x_i \in \{0, 1\}$ is relaxed to a continuous counterpart $p_i \in [0, 1]$, and $\hat{f} : [0, 1]^N \times \mathcal{C} \to \mathbb{R}$ denotes the relaxed form of $f$ such that $\hat{f}(\boldsymbol{x}; C) = f(\boldsymbol{x}; C)$ for $\boldsymbol{x} \in \{0, 1\}^N$. The relation between each constraint $v_i$ and its relaxation $\hat{v}_i$ is similar for $i \in [I+J]$, meaning that $\forall i \in [I+J], \ \hat{v}_i(\boldsymbol{x}; C) = v_i(\boldsymbol{x}; C)$ for $\boldsymbol{x} \in \{0, 1\}^N$. Wang et al. [2022] and Schuetz et al. [2022a] formulated $\mathcal{A}_\theta(C)$ as the relaxed continuous variables, defined as $\mathcal{A}_\theta(\cdot) : \mathcal{C} \to [0, 1]^n$. In the following discussions, we denote

$\mathcal{A}_\theta$ as $p_\theta$ to make the parametrization of the relaxed variables explicit. Then, $p_\theta$ is optimized by directly minimizing the following label-independent function:

$$\hat{l}(\boldsymbol{\theta}; C, \boldsymbol{\lambda}) \triangleq \hat{f}(\boldsymbol{p}_\theta(C); C) + \sum_{i=1}^{I+J} \lambda_i \hat{v}_i(\boldsymbol{p}_\theta(C); C).$$

After training, the relaxed solution $p_\theta$ is converted into discrete variables using artificial rounding $p_{\boldsymbol{\theta}}$, where $\forall i \in [N]$, $x_i = \text{int}(p_{\boldsymbol{\theta},i}(C))$ based on a threshold [Schuetz et al., 2022a], or alternatively, a greedy method [Wang et al., 2022]. Two types of schemes for UL-based solvers have been developed based on this formulation.

**(Type I) Learning generalized heuristics from history/data.** One approach, proposed by Karalias and Loukas [2020], aims to automatically learn effective heuristics from historical dataset instances $\mathcal{D} = \{C_\mu\}_{\mu=1}^P$ and then apply these learned heuristics to a new instance $C^*$, through inference. Note that this method assumes that either the training dataset is easily obtainable or that meaningful data augmentation is feasible. Specifically, given a set of training instances $\mathcal{D} = (C_\mu)$, sampled independently and identically from a distribution $P(C)$, the goal is to minimize the average loss function $\min_{\boldsymbol{\theta}} \sum_{\mu=1}^P l(\boldsymbol{\theta}; C_\mu, \boldsymbol{\lambda})$. However, this method does not guarantee quality for a test instance, $C^*$. Even if the training instances $\mathcal{D}$ are extensive and the test instance $C$ follows $P(C)$, low average performance $\mathbb{E}_{C \sim P(C)}[\hat{l}(\theta; C)]$ may not guarantee a low $l(\theta; C)$ for on a specific $C$. To address this issue, Wang and Li [2023] introduced a meta-learning approach where NNs aim to provide good initialization for future instances rather than direct solutions.

**(Type II) Learning effective heuristics on a specific single instance.** Another approach, known as the PI-GNN solver [Schuetz et al., 2022a,b], automatically learns instance-specific heuristics for a single instance using the instance parameter $C$ by directly applying Eq. (2.1). This approach addresses CO problems on graphs, where $C = G(V, E)$, and employs GNNs for the relaxed variables $p_\theta(G)$. Here, an $L$-layered GNN is trained to directly minimize $\hat{l}(\boldsymbol{\theta}; C, \boldsymbol{\lambda})$, taking as input a graph $G$ and the embedding vectors on its nodes, and outputting the relaxed solution $\boldsymbol{p}_\theta(G) \in [0, 1]^N$. A detailed description of GNNs is provided in Appendix E.2. Note that this setting is applicable even when the training dataset $\mathcal{D}$ is difficult to obtain. The overparameterization of relaxed variables is expected to smooth the objective function by introducing additional parameters to the optimization problem, similar to the kernel method. However, minimizing Eq. 2.1 for a single instance can be time-consuming compared to the inference process. Nonetheless, for large-scale CO problems, this approach has been reported to outperform other solvers in terms of both computational time and solution quality [Schuetz et al., 2022a,b].

Note that, while both UL-based solvers for multiple instances (Type I) and individual instances (Type II) are valuable, this study focuses on advancing the latter: a UL-based solver for a single instance. Both types of solvers are applicable to cost functions that meet a particular requirement due to their reliance on a gradient-based algorithm to minimize Eq (2.1).

**Assumption 2.1** (Differentiable cost function). The relaxed loss function $\hat{l}(\boldsymbol{\theta}; C, \boldsymbol{\lambda})$ and its partial derivative $\partial \hat{l}(\boldsymbol{\theta}; C, \boldsymbol{\lambda})/\partial \boldsymbol{\theta}$ are accessible during the optimization process.

These requirements encompass a nonlinear cost function and interactions involving many-body interactions, extending beyond simple two-body interactions.

# 3 Continuous relaxation annealing for UL-based solvers

In this section, we discuss the practical issues associated with UL-based solvers and then introduce continuous relaxation annealing (CRA) as a proposed solution.

## 3.1 Motivation: practical issues of UL-based solvers

UL-based solvers (Type II) [Schuetz et al., 2022a,b] are effective in addressing large-scale CO problems. However, these solvers present following two practical issues, highlighted in several recent studies [Wang and Li, 2023, Angelini and Ricci-Tersenghi, 2023]. Additionally, we numerically validate these issues; see Appendix F.1 for detailed results.

**(I) Ambiguity in rounding method after learning.** UL-based solvers employ a continuous relaxation strategy to train NNs and then convert the relaxed continuous variables into discrete binary values through artificial rounding as discussed in Section 2.1. This inherent ambiguity in continuous relaxation strategy often results in potential discrepancies between the optimal solutions of the original discrete CO problem and those of the relaxed continuous one. Continuous relaxation expands the solution space, often producing continuous values that lower the cost compared to an optimal binary value. Indeed, while linear relaxation can provide an optimal solution for discrete problems on bipartite graphs [Hoffman and Kruskal, 2010], it typically results in solutions with 1/2 values, which is known to half-integrality [Nemhauser and Trotter Jr, 1974]. Existing rounding methods [Schuetz et al., 2022b, Wang et al., 2022] often lose robustness in these scenarios. In practice, PI-GNN solver often outputs values near 1/2, underscoring the limitations of current rounding techniques for UL-based solvers.

**(II) Difficulty in optimizing NNs.** Recently, Angelini and Ricci-Tersenghi [2023] demonstrated that PI-GNN solver falls short of achieving results comparable to those of the degree-based greedy algorithm (DGA) [Angelini and Ricci-Tersenghi, 2019] when solving the MIS problems on RRGs. Angelini and Ricci-Tersenghi [2023] further emphasized the importance of evaluating UL-based solvers on complex CO problems, where greedy algorithms typically perform worse. A representative example is the MIS problems on RRGs with a constant degree $d > 16$, where a clustering transition in the solution space creates barriers that impede optimization. Moreover, Wang and Li [2023] emphasized the importance of using training/historical datasets, $\mathcal{D} = \{C_\mu\}_{1 \leq \mu \leq P}$, which contain various graphs and initialization using outputs from greedy solvers, such as DGA and RGA for MIS problems. Their numerical analysis indicated that PI-GNN solver tends to get trapped in local optima when directly optimized directly for a single instance without leveraging a training dataset $\mathcal{D}$. However, in a practical setting, systematic methods for generating or collecting training datasets $\mathcal{D}$ to effectively avoid local optima remains unclear. Additionally, training on instances that do not contribute to escaping local optima is time-consuming. Therefore, it is crucial to develop an effective UL-based solver that can operate on a single instance without relying on training data, $\mathcal{D}$. Our numerical experiments, detailed in Appendix F.1, also confirmed this optimization issue. They demonstrated that as problem complexity increases, the PI-GNN solver is often drawn into trivial local optima, $\boldsymbol{p_\theta} = \boldsymbol{0}_N$, in certain problems. This entrapment results in prolonged plateaus that significantly slow down the learning process and, in especially challenging cases, can render learning entirely infeasible. Our numerical experiments, detailed in Appendix F.1, also validated this optimization issue, demonstrating that as the problem complexity increases, PI-GNN solver tends to be absorbed into the trivial local optima $\boldsymbol{p_\theta} = \boldsymbol{0}_N$ in some problems, resulting in prolonged plateaus which significantly decelerates the learning process and, in particularly challenging cases, can render learning entirely infeasible.

### 3.2 Continuous relaxation annealing

**Penalty term to control discreteness and continuity.** To address these issues, we propose a penalty term to control the balance between discreteness and continuity in the relaxed variables, formulated as follows:

$$\hat{r}(\boldsymbol{p}; C, \boldsymbol{\lambda}, \gamma) = \hat{l}(\boldsymbol{p}; C, \boldsymbol{\lambda}) + \gamma\Phi(\boldsymbol{p}), \ \ \Phi(\boldsymbol{p}) \triangleq \sum_{i=1}^{N}(1 - (2p_i - 1)^\alpha), \ \ \alpha \in \{2n \mid n \in \mathbb{N}_+\},$$

where $\gamma \in \mathbb{R}$ is a penalty parameter, and the even number $\alpha$ denote a curve rate. When $\gamma$ is negative, i.e., $\gamma < 0$, the relaxed variables tend to favor the continuous space, smoothing the non-convexity of the objective function $\hat{l}(\boldsymbol{p}; C, \boldsymbol{\lambda})$ due to the convexity of the penalty term $\Phi(\boldsymbol{p})$. In contrast, when $\gamma$ is positive, i.e., $\gamma > 0$, the relaxed variables tend to favor discrete space, smoothing out the continuous solution into discrete solution. Formally, the following theorem holds as $\lambda$ approaches $\pm\infty$.

**Theorem 3.1.** *Assuming the objective function $\hat{l}(\boldsymbol{p}; C)$ is bounded within the domain $[0, 1]^N$, as $\gamma \to +\infty$, the relaxed solutions $\boldsymbol{p}^* \in \arg\min_{\boldsymbol{p}} \hat{r}(\boldsymbol{p}; C, \boldsymbol{\lambda}, \gamma)$ converge to the original solutions $\boldsymbol{x}^* \in \arg\min_{\boldsymbol{x}} l(\boldsymbol{x}; C, \boldsymbol{\lambda})$. Moreover, as $\gamma \to -\infty$, the loss function $\hat{r}(\boldsymbol{p}; C, \boldsymbol{\lambda}, \gamma)$ becomes convex, and the relaxed solution $\boldsymbol{1}_N/2 = \arg\min_{\boldsymbol{p}} \hat{r}(\boldsymbol{p}, C, \boldsymbol{\lambda}, \gamma)$ is unique.*

For the detailed proof, refer to Appendix B.1. Theorem 3.1 can be generalized for any convex function $\Phi(\boldsymbol{p}; C)$ that has a unique maximum at $\boldsymbol{1}_N/2$ and achieves a global minimum for all $\boldsymbol{p} \in \{0, 1\}^N$; an

example is binary cross entropy $\Phi_{\mathrm{CE}}(\boldsymbol{p}) = \sum_{i=1}^{N}(p_i \log p_i + (1-p_i)\log(1-p_i))$, introduced by Sun et al. [2022], Sanokowski et al. [2024] for the UL-based solvers (Type I). Additionally, the penalty term eliminates the stationary point $\boldsymbol{p}^* = \boldsymbol{0}_N$ described in Section 3.1, preventing convergence to a plateau. For UL-based solvers, the penalty term is expressed as follows:

$$\hat{r}(\boldsymbol{\theta}; C, \boldsymbol{\lambda}, \gamma) = \hat{l}(\boldsymbol{\theta}; C, \boldsymbol{\lambda}) + \gamma \Phi(\boldsymbol{\theta}; C),$$

where $\Phi(\boldsymbol{\theta}; C) \triangleq \Phi(\boldsymbol{p}_\theta(C))$. According to Theorem 3.1, setting a sufficiently large $\gamma$ value cases the relaxed variables to approach nearly discrete values. We can also generalize this penalty term $\Phi(\boldsymbol{\theta}; C)$, to Potts variables optimization, including coloring problems [Schuetz et al., 2022b], and mixed-integer optimization; refer to Appendix C.1.

**Annealing penalty term.** We propose an annealing strategy that gradually anneals the penalty parameter $\gamma$ in Eq. (3.2). Initially, a negative gamma value, i.e., $\gamma < 0$, is chosen to leverage the properties, facilitating broad exploration by smoothing the non-convexity of $\hat{l}(\boldsymbol{\theta}; C, \boldsymbol{\lambda})$ and eliminating the stationary point $\boldsymbol{p}^* = \boldsymbol{0}_N$ to avoid the plateau, as discussed in Section 3.1. Subsequently, the penalty parameter $\gamma$ is gradually increased to a positive value, $\gamma > 0$, with each update of the trainable parameters (one epoch), until the penalty term approaches zero, i.e., $\Phi(\boldsymbol{\theta}, C) \approx 0$, to automatically round the relaxed variables by smoothing out suboptimal continuous solutions oscillating between 1 or 0. A conceptual diagram of this annealing process is shown in Fig. 1.

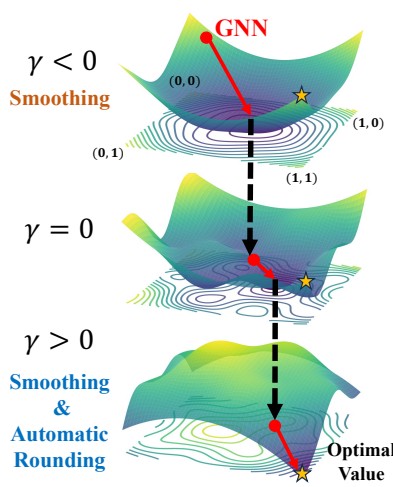

Figure 1: Annealing strategy. When $\gamma < 0$, it facilitates exploration by reducing the non-convexity of the objective function. As $\gamma$ increases, it promotes optimal discrete solutions by smoothing away suboptimal continuous ones.

Note that employing the binary cross-entropy $\Phi_{\mathrm{CE}}(\boldsymbol{p})$ is infeasible for UL-based solvers when $\gamma > 0$, as the gradient $\partial\Phi_{\mathrm{CE}}(\boldsymbol{p})/\partial p_i$ diverges to $\pm\infty$ at 0 or 1. In deed, when $\gamma = 0$, most relaxed variables typically approach binary values, with a relatively small number of variables oscillating between 0 and 1. This gradient divergence issue in $\Phi_{\mathrm{CE}}(\boldsymbol{p})$ makes the learning infeasible without additional techniques, such as gradient clipping. In contrast, the gradient of the penalty term in Eq. 3.2, $\partial\Phi(\boldsymbol{p})/\partial p_i$, is bounded within $[-2\alpha, 2\alpha]$ for any $\gamma$, preventing the gradient divergence issue seen in $\Phi_{\mathrm{CE}}(\boldsymbol{p})$. Additionally, by increasing $\alpha$, the absolute value of the gradient near $1/2$ becomes smaller, allowing for control over the smoothing strength toward a discrete solution near $1/2$.

We also propose an early stopping strategy that monitors both the loss function and the penalty term, halting the annealing and learning processes when the penalty term approaches zero, i.e., $\Phi(\boldsymbol{\theta}; C) \approx 0$. Various annealing schedules can be considered; in this study, we employ the following scheduling: $\gamma(\tau + 1) \leftarrow \gamma(\tau) + \varepsilon$, where the scheduling rate $\varepsilon \in \mathbb{R}_+$ is a small constant, and $\tau$ denotes the update iterations of the trainable parameters. We refer to the PI-GNN solver with this continuous relaxation annealing as CRA-PI-GNN solver. Here, two additional hyperparameters are introduced: the initial scheduling value $\gamma(0)$ and the scheduling rate $\varepsilon$. Numerical experiments suggest that better solutions are obtained when $\gamma(0)$ is set to a small negative value and $\varepsilon$ is kept low. The ablation study are presented in Appendix F.5.

## 4   Related Work

Previous works on UL-based solvers have addressed various problems, such as MaxCut problems [Yao et al., 2019] and traveling salesman problems [Hudson et al., 2021], using carefully tailored problem-specific objectives. Some studies have also explored constraint satisfaction problems [Amizadeh et al., 2018, Toenshoff et al., 2019], but applying these approaches to broader CO problems often requires problem-specific reductions. Karalias and Loukas [2020] proposed Erdős Goes Neural (EGN) solver, an UL-based solver for general CO problems based on Erdős' probabilistic method. This solver

generate solutions through an inference process using training instances. Subsequently, Wang et al. [2022] proposed an entry-wise concave continuous relaxation, broadening the EGN solver to a wide range of CO problems. In contrast, Schuetz et al. [2022a,b] proposed PI-GNN solver, an UL-based solver for a single CO problems that automatically learns problem-specific heuristics during the training process. However, Angelini and Ricci-Tersenghi [2023], Boettcher [2023] pointed out the optimization difficulties where PI-GNN solver failed to achieve results comparable to those of greedy algorithms. Wang and Li [2023] also claimed optimization issues with PI-GNN solver, emphasizing the importance of learning from training data and history to overcome local optima. They then proposed Meta-EGN solvers, a meta-learning approach that updates NN network parameters for individual CO problem instances. Furthermore, to address these optimization issue, Lin et al. [2023], Sun et al. [2022], Sanokowski et al. [2024] proposed annealing strategy similar to simulated annealing [Kirkpatrick et al., 1983].

# 5 Experiments

We begin by evaluating the performance of CRA-PI-GNN solver on the MIS and the MaxCut benchmark problems across multiple graphs of varying sizes, demonstrating that CRA effectively overcomes optimization challenges without relying on data/history $\mathcal{D}$. We then extend the evaluation to the DBM problems, showing the applicability to more practical CO problems. For the objective functions and the detailed explanations, refer to Appendix E.1.

## 5.1 Experimental settings

**Baseline methods.**  In all experiments, the baseline methods include the PI-GNN solver [Schuetz et al., 2022a] as the direct baseline of a UL-based solver for a single instance. For the MIS problems, we also consider the random greedy algorithm (RGA) and DGA [Angelini and Ricci-Tersenghi, 2019] as heuristic baselines. For the MaxCut problems, RUN-CSP solver [Toenshoff et al., 2019] is considered as an additional baseline, and a standard greedy algorithm and SDP based approximation algorithm [Goemans and Williamson, 1995] are considered as an additional classical baseline. The parameters for the Goemans-Williamson (GW) approximation are all set according to the settings in Schuetz et al. [2022b]. The implementation used the open-source CVXOPT solver with CVXPY [2] as the modeling interface. Note that we do not consider UL-based solvers for learning generalized heuristics [Karalias and Loukas, 2020, Wang et al., 2022, Wang and Li, 2023], which rely on training instances $\mathcal{D} = \{C_\mu\}_{\mu=1}^{P}$. The primary objective of this study is to evaluate whether CRA-PI-GNN solver can surpass the performance of both PI-GNN solver and greedy algorithms. However, for the MIS problem, EGN solver [Karalias and Loukas, 2020] and Meta-EGN solver [Wang and Li, 2023] are considered to confirm that CRA can overcome the optimization issues without training instances.

**Implementation.**  The objective of the numerical experiments is to compare the CRA-PI-GNN solver with the PI-GNN solver. Thus, we follow the same experimental configuration described as the experiments in Schuetz et al. [2022a], employing a simple two-layer `GCV` and `GraphSAGE` [Hamilton et al., 2017] implemented by the Deep Graph Library [Wang et al., 2019]; Refer to Appendix D.1 for the detailed architectures. We use the AdamW [Kingma and Ba, 2014] optimizer with a learning rate of $\eta = 10^{-4}$ and weight decay of $10^{-2}$. The GNNs are trained for up to $5 \times 10^4$ epochs with early stopping, which monitors the summarized loss function $\sum_{s=1}^{S} \hat{l}(P_{:,s})$ and the entropy term $\Phi(P; \gamma, \alpha)$ with tolerance $10^{-5}$ and patience $10^3$ epochs; Further details are provided in Appendix D.2. We set the initial scheduling value to $\gamma(0) = -20$ for the MIS and matching problems, and we set $\gamma(0) = -6$ for the MaxCut problems with the scheduling rate $\varepsilon = 10^{-3}$ and curve rate $\alpha = 2$ in Eq. (3.2). These values are not necessarily optimal, and refining these parameters can lead to better solutions; Refer to Appendix F.5 and Appendix F.6 for an ablation study of these parameters. Once the training process is complete, we apply projection heuristics to map the obtained soft solutions back to discrete solutions using simple projection, where for all $i \in [N]$, we map $p_{\theta,i}$ into 0 if $p_{\theta,i} \leq 0.5$ and $p_{\theta,i}$ into 1 if $p_{\theta,i} > 0.5$. However, due to the early stopping in Section 3.2, the CRA-PI-GNN solver ensures that for all benchmark CO problems, the soft solution at the end of the training process became 0 or 1 within the 32-bit Floating Point range in Pytorch GPU; thus, it is robust against a given threshold, which we set to 0.5 in our experiments. Additionally, no violations

---

[2] https://github.com/hermish/cvx-graph-algorithms.

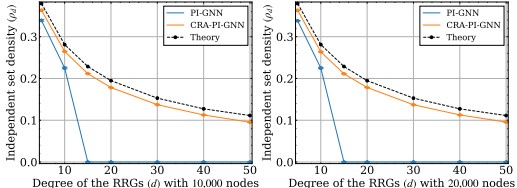
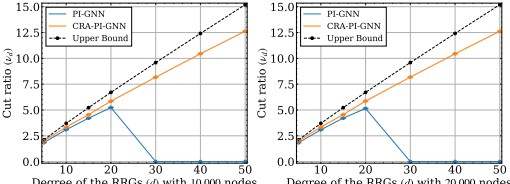

Figure 2: Independent set density of the MIS problem on $d$-RRG. Results for graphs with $N = 10,000$ nodes (Left) and $N = 20,000$ nodes (Right). the dashed lines represent the theoretical results [Barbier et al., 2013].

Figure 3: Cut ratio of the MaxCut problem on $d$-RRG as a function of the degree $d$ Results for $N = 10,000$ (Left) and $N = 20,000$ (Right). The dashed lines represents the theoretical upper bounds [Parisi, 1980].

Table 1: ApR in MIS problems on RRGs with 10,000 nodes and node degree $d = 20, 100$. "CRA" represents the CRA-PI-GNN solver.

| Instance | Method | ApR |
|---|---|---|
| 20-RRG | RGA | $0.776 \pm 0.001$ |
| | DGA | $0.891 \pm 0.001$ |
| | EGN | $0.775$ (2023) |
| | META-EGN | $0.887$ (2023) |
| | PI-GNN (GCV) | $0.000 \pm 0.000$ |
| | PI-GNN (SAGE) | $0.745 \pm 0.003$ |
| | **CRA (GCV)** | $0.937 \pm 0.002$ |
| | **CRA (SAGE)** | $\mathbf{0.963 \pm 0.001}$ |
| 100-RRG | RGA | $0.663 \pm 0.001$ |
| | DGA | $0.848 \pm 0.002$ |
| | PI-GNN (GCV) | $0.000 \pm 0.000$ |
| | PI-GNN (SAGE) | $0.000 \pm 0.000$ |
| | **CRA (GCV)** | $0.855 \pm 0.004$ |
| | **CRA (SAGE)** | $\mathbf{0.924 \pm 0.001}$ |

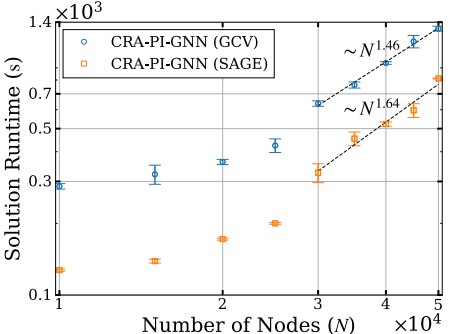

Figure 4: (Right) computational runtime (in seconds) of the CRA-PI-GNN solvers with the GraphSage and Conv architectures on 100-RRG with varying numbers of nodes $N$. Error bars represent the standard deviations of the results.

of the constraints were observed in our numerical experiments. Thus, following results presented in are feasible solutions.

**Evaluation metrics.** Following Karalias and Loukas [2020], We use the approximation ratio (ApR), formulated as $\mathrm{ApR} = f(\boldsymbol{x}; C)/f(\boldsymbol{x}^*; C)$, where $\boldsymbol{x}^*$ is optimal solution. For the MIS problems, we evaluate the ApRs using the theoretical optimal cost [Barbier et al., 2013] and the independent set density $\rho$ relative to the theoretical results. For the MaxCut problems on RRGs, we adopt the cut ratio $\nu$ against the theoretical upper bound [Parisi, 1980, Dembo et al., 2017]; see Appendix E.1 for the details. All the results for the MIS and MaxCut problems are summarized based on 5 RRGs with different random seeds. In the case of the MaxCut Gset problem, the ApR is calculated compared to the known best cost functions. Regarding the DBM problems, we calculate the ApR against the global optimal, identified using Gurobi 10.0.1 solver with default settings.

## 5.2 MIS problems

**Degree dependency of solutions using CRA.** First, we compare the performance of the PI-GNN and CRA-PI-GNN solvers using GCV, as in Schuetz et al. [2022a]. Fig. 2 shows the independent set density $\rho_d$ as a function of degree $d$ obtained by the PI-GNN and CRA-PI-GNN solvers compared with the theoretical results [Barbier et al., 2013]. Across all degrees $d$, the CRA-PI-GNN solver outperformed the PI-GNN solver and approached the theoretical results, whereas the PI-GNN solver fail to find valid solutions, especially for $d \geq 15$, as pointed by the previous studies [Angelini and Ricci-Tersenghi, 2023, Wang and Li, 2023].

**Response to Angelini and Ricci-Tersenghi [2023] and Wang and Li [2023].** MIS problems on RRGs with a degree $d$ larger than 16 is known to be hard problems [Barbier et al., 2013]. As

discussed in Section 3.1, Angelini and Ricci-Tersenghi [2023], Wang and Li [2023] have posted the optimization concerns on UL-based solvers. However, we call these claim into question by substantially outperforming heuristics DGA and RGA for the MIS on graphs with $d = 20, 100$, without training/historical instances $\mathcal{D} = \{G^\mu\}_{\mu=1}^n$, as shown in Table 1. See Appendix 6 for the results of solving all other Gsets, where consistently, CRA-PI-GNN provides better results as well. A comparison of the sampling-based solvers, RL-based solvers, SL-based solvers, Gurobi, and MIS-specific solvers is presented in Appendix F.2.

**Acceleration of learning speed.** We also compared the learning curve between PI-GNN and CRA-PI-GNN solver to confirmed that the CRA-PI-GNN solver does not become trapped in the plateau, $\boldsymbol{p}_N = \boldsymbol{0}_N$, as discussed in Section 3.1. Fig. 5 shows the dynamics of the cost functions for the MIS problems with $N = 10,000$ across $d = 3, 5, 20, 100$. Across all degrees, CRA-PI-GNN solver achieves a better solution with fewer epochs than PI-GNN solver. Specifically, PI-GNN solver becomes significantly slower due to getting trapped in the plateau even for graphs with low degrees, such as $d = 3, 5$. In contrast, CRA-PI-GNN solver can effectively escape from plateaus through the smoothing and automatic rounding of the penalty term when the negative parameter $\gamma > 0$.

**Computational scaling.** We next evaluate the computational scaling of the CRA-PI-GNN solver for MIS problems with large-scale RRGs with a node degree of 100 in Fig. 4, following previous studies [Schuetz et al., 2022a, Wang and Li, 2023]. Fig. 4 demonstrated a moderate super-linear scaling of the total computational time, approximately $\sim N^{1.4}$ for GCN and $\sim N^{1.7}$ for GraphSage. This performance is nearly identical to that of the PI-GNN solver [Schuetz et al., 2022a] for problems on RRGs with lower degrees. It is important note that the runtimes of CRA-PI-GNN solver heavily depend on the optimizer for GNNs and annealing rate $\varepsilon$; thus this scaling remains largely unchanged for problems other than the MIS on 100 RRG. Additionally, CRA demonstrate that the

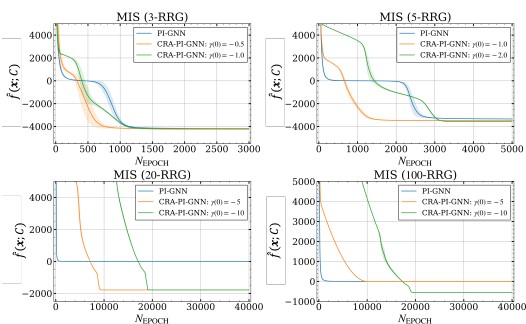

Figure 5: The dynamics of cost function for MIS problems on RRGs with $N = 10,000$ nodes varying degrees $d$ as a function of the number of parameters updates $N_{\text{EPOCH}}$.

runtime remains nearly constant as graph order and density increase, indicating effective scalability with denser graphs which is presented in Appendix F.2.

## 5.3 MaxCut problem

**Degree dependency of solutions.** We first compare the performances of PI-GNN and CRA-PI-GNN solvers with GCV, following Schuetz et al. [2022a]. Fig. 3 shows the cut ratio $\nu_d$ as a function of the degree $d$ compared to the theoretical upper bound [Parisi, 1980, Dembo et al., 2017]. Across all degrees $d$, CRA-PI-GNN solver also outperforms PI-GNN solver, approaching the theoretical upper bound. In contrast, PI-GNN solver fails to find valid solutions for $d > 20$ as with the case of the MIS problems in Section 5.2.

**Standard MaxCut benchmark test.** Following Schuetz et al. [2022a], we next conducted additional experiments on standard MaxCut benchmark instances based on the publicly available Gset dataset [Ye, 2003], which is commonly used to evaluate MaxCut algorithms. Here, we provide benchmark results for seven distinct graphs with thousands of nodes, including Erdös-Renyi graphs with uniform edge probability, graphs in which the connectivity decays gradually from node 1 to $N$, 4-regular toroidal graphs, and a very large Gset instance with $N = 10,000$ nodes. Table 2 shows, across all problems, CRA-PI-GNN solver outperforms both the PI-GNN, RUN-CSP solvers and other greedy algorithm.

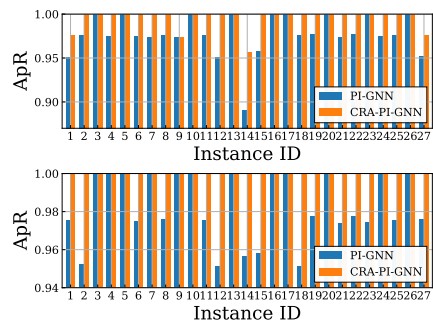

Figure 6: ApR on DBM problems.

Table 2: ApR for MaxCut on Gset

| GRAPH | (NODES, EDGES) | GREEDY | SDP | RUN-CSP | PI-GNN | **CRA** |
|-------|----------------|--------|-----|---------|--------|---------|
| G14 | (800, 4,694) | 0.946 | 0.970 | 0.960 | 0.988 | **0.994** |
| G15 | (800, 4,661) | 0.939 | 0.958 | 0.960 | 0.980 | **0.992** |
| G22 | (2,000, 19,990) | 0.923 | 0.77 | 0.975 | 0.987 | **0.998** |
| G49 | (3,000, 6,000) | **1.000** | **1.000** | **1.000** | 0.986 | **1.000** |
| G50 | (3,000, 6,000) | **1.000** | **1.000** | **1.000** | 0.990 | **1.000** |
| G55 | (5,000, 12,468) | 0.892 | – | 0.982 | 0.983 | **0.991** |
| G70 | (10,000, 9,999) | 0.886 | – | 0.970 | 0.982 | **0.992** |

See Appendix 6 for the results of solving all other Gsets, where CRA-PI-GNN consistently provides better results as well.

### 5.4 Diverse bipartite matching

To evaluate the applicability of the CRA-PI-GNN solver to more practical problems not on graphs, we conducted experiments on DBM problems [Ferber et al., 2020, Mulamba et al., 2020, Mandi et al., 2022]; refer to Appendix E.1 for details. This problems consists of 27 distinct instances with varying properties, and each instance comprises 100 nodes representing scientific publications, divided into two groups of 50 nodes $N_1$ and $N_2$. The optimization is formulated as follows:

$$l(\boldsymbol{x}; C, M, \boldsymbol{\lambda}) = -\sum_{ij} C_{ij} x_{ij} + \lambda_1 \sum_i \mathrm{ReLU}\left(\sum_j x_{ij} - 1\right) + \lambda_2 \sum_j \mathrm{ReLU}\left(\sum_i x_{ij} - 1\right)$$

$$+ \lambda_3 \mathrm{ReLU}\left(p \sum_{ij} x_{ij} - \sum_{ij} M_{ij} x_{ij}\right) + \lambda_4 \mathrm{ReLU}\left(q \sum_{ij} x_{ij} - \sum_{ij} (1 - M_{ij}) x_{ij}\right),$$

where $C \in \mathbb{R}^{N_1 \times N_2}$ represents the likelihood of a link between each pair of nodes, an indicator $M_{ij}$ is set to 0 if article $i$ and $j$ share the same subject field (1 otherwise) $\forall i \in N_1$, and $j \in N_2$. The parameters $p, q \in [0, 1]$ represent the probability of pairs sharing their field and of unrelated pairs, respectively. As in Mandi et al. [2022], we explore two variations of this problem, with $p = q =$ being 25% and 5%, respectively, and these variations are referred to as Matching-1 and Matching-2, respectively. In this experiment, we set $\lambda_1 = \lambda_2 = 10$ and $\lambda_3 = \lambda_4 = 25$. Fig 6 shows that the CRA-PI-GNN solver can find better solutions across all instances.

## 6   Conclusion

This study proposes CRA strategy to address the both optimization and rounding issue in UL-based solvers. CRA strategy introduces a penalty term that dynamically shifts from prioritizing continuous solutions, where the non-convexity of the objective function is effectively smoothed, to enforcing discreteness, thereby eliminating artificial rounding. Experimental results demonstrate that CRA-PI-GNN solver significantly outperforms PI-GNN solver and greedy algorithms across various complex CO problems, including MIS, MaxCut, and DBM problems. CRA approach not only enhances solution quality but also accelerates the learning process.

**Limitation.** In these numerical experiments, most hyperparameters were fixed to their default values, as outlined in Section 5.1, with minimal tuning. However, tuning may be necessary for specific problems or to further enhance performance.

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

# A   Overview

This supplementary material provides extended explanations, implementation details, and additional results.

# B   Derivation

## B.1   Proof of Theorem 3.1

First, we present three lemmas, and then we demonstrate Theorem 3.1 based on these lemmas.

**Lemma B.1.** *For any even natural number $\alpha \in \{2n \mid n \in \mathbb{N}_+\}$, the function $\phi(p) = 1 - (2p - 1)^\alpha$ defined on $[0, 1]$ achieves its maximum value of $1$ when $p = 1/2$ and its minimum value of $0$ when $p = 0$ or $p = 1$.*

*Proof.* The derivative of $\phi(p)$ relative to $p$ is $d\phi(p)/dp = -2\alpha(2p - 1)$, which is zero when $p = 1/2$. This is a point where the function is maximized because the second derivative $d^2\phi(p)/dp^2 = -4\alpha \leq 0$. In addition, this function is concave and symmetric relative to $p = 1/2$ because $\alpha$ is an even natural number, i.e., $\phi(p) = \phi(1 - p)$, thereby achieving its minimum value of $0$ when $p = 0$ or $p = 1$.   □

**Lemma B.2.** *For any even natural number $\alpha \in \{2n \mid n \in \mathbb{N}_+\}$, if $\gamma \to +\infty$, minimizing the penalty term $\gamma\Phi(\boldsymbol{p}) = \gamma \sum_{i=1}^N (1 - (2p_i - 1)^\alpha) = \gamma \sum_{i=1}^N \phi(p_i)$ enforces that the for all $i \in [N]$, $p_i$ is either $0$ or $1$ and, if $\gamma \to -\infty$, the penalty term enforces $\boldsymbol{p} = \mathbf{1}_N/2$.*

*Proof.* From Lemma B.1, as $\gamma \to +\infty$, $\phi(p)$ is minimal value when, for any $i \in [N]$, $p_i = 0$ or $p_i = 1$. As $\gamma \to -\infty$, $\phi(p; \alpha, \gamma)$ is minimal value when, for any $i \in [N]$, $p_i = 1/2$.   □

**Lemma B.3.** *For any even number $\alpha \in \{2n \mid n \in \mathbb{N}_+\}$, $\gamma\Phi(\boldsymbol{p})$ is concave when $\lambda > 0$ and is a convex function when $\lambda < 0$.*

*Proof.* Note that $\gamma\Phi(\boldsymbol{p}) = \gamma \sum_{i=1}^N \phi(p_i) = \gamma \sum_{i=1}^N (1 - (2p_i - 1)^\alpha)$ is separable across its components $p_i$. Thus, it is sufficient to prove that each $\gamma\phi_i(p_i; \alpha)$ is concave or convex in $p_i$ because the sum of the concave or convex functions is also concave (and vice versa). Thus, we consider the second derivative of $\gamma\phi_i(p_i)$ with respect to $p_i$:

$$\gamma \frac{d^2\phi_i(p_i)}{dp_i^2} = -4\gamma\alpha.$$

If $\gamma > 0$, the second derivative is negative for all $p_i \in [0, 1]$, and this completes the proof that $\gamma\Phi(\boldsymbol{p})$ is a concave function when $\gamma$ is positive (and vice versa).   □

Combining Lemma B.1, Lemma B.2 and Lemma B.3, one can show the following theorem.

**Theorem B.4.** *Under the assumption that the objective function $\hat{l}(\boldsymbol{p}; C)$ is bounded within the domain $[0, 1]^N$, as $\gamma \to +\infty$, the soft solutions $\boldsymbol{p}^* \in \arg\min_{\boldsymbol{p}} \hat{r}(\boldsymbol{p}; C, \boldsymbol{\lambda}, \gamma)$ converge to the original solutions $\boldsymbol{x}^* \in \arg\min_{\boldsymbol{x}} l(\boldsymbol{x}; C, \boldsymbol{\lambda})$. In addition, as $\gamma \to -\infty$, the loss function $\hat{r}(\boldsymbol{p}; C, \boldsymbol{\lambda}, \gamma)$ becomes convex, and the soft solution $\mathbf{1}_N/2 = \arg\min_{\boldsymbol{p}} \hat{r}(\boldsymbol{p}, C, \boldsymbol{\lambda}, \gamma)$ is unique.*

*Proof.* As $\lambda \to +\infty$, the penalty term $\Phi(\boldsymbol{p})$ dominates the loss function $\hat{r}(\boldsymbol{p}; C, \boldsymbol{\lambda}, \gamma)$. According to Lemma B.2, this penalty term forces the optimal solution $\boldsymbol{p}^*$ to a binary vector whose components, for all $i \in [N]$ $p_i^*$ that are either $0$ or $1$ because any non-binary value results in an infinitely large penalty. This effectively restricts the feasible region to the vertices of the unit hypercube, which correspond to the binary vector in $\{0, 1\}^N$. Thus, as $\lambda \to \infty$, the solutions to the relaxed problem converge to those of the original problem. As $\lambda \to -\infty$, the penalty term $\Phi(\boldsymbol{p})$ also dominates the loss function $\hat{r}(\boldsymbol{p}; C, \boldsymbol{\lambda}, \gamma)$ and the $\hat{r}(\boldsymbol{p}; C, \boldsymbol{\lambda})$ convex function from Lemma B.3. According to Lemma B.2, this penalty term forces the optimal solution $\boldsymbol{p}^* = \mathbf{1}_N/2$.   □

The theorem holds for the cross entropy penalty given by

$$\Phi(\boldsymbol{p}) = \sum_{i=1}^{N} \left( p_i \log(p_i) + (1 - p_i) \log(1 - p_i) \right)$$

in the UL-based solver using data or history [Sun et al., 2022, Sanokowski et al., 2024] because $\Phi(\boldsymbol{p})$ can similarly satisfy Lemma B.1, Lemma B.2 and Lemma B.3.

**Corollary B.5.** *Theorem B.4 holds for the following penalty term:*

$$\Phi(\boldsymbol{p}) = \sum_{i=1}^{N} \left( p_i \log(p_i) + (1 - p_i) \log(1 - p_i) \right).$$

## C   Generalization of CRA

### C.1   Generalization for to Potts variable optimization

This section generalize the penalty term $\Phi(\boldsymbol{\theta}; C)$ introduced for binary variables to $K$-Potts variables. $K$-Potts variable is the Kronecker delta $\delta(x_i, x_j)$ which equas one whenever $x_i = x_j$ and zero otherwise and a decision variable takes on $K$ different values, $\forall i \in [N]$, $x_i = 1, \ldots, K$. For example, graph $K$-coloring problems can be expressed as

$$f(\boldsymbol{x}; G(V, E)) = - \sum_{(i,j) \in E} \delta(x_i, x_j).$$

For Potts variables, the output of the GNN is $\boldsymbol{h}_{\boldsymbol{\theta}}^L = P(\boldsymbol{\theta}; C) \in \mathbb{R}^{N \times K}$ and the penalty term can be generalized as follows:

$$\Phi(\boldsymbol{\theta}; C) = \sum_{i=1}^{N} \left( 1 - \sum_{k=1}^{K} (K P_{i,k}(\boldsymbol{\theta}; C) - 1)^{\alpha} \right).$$

## D   Additional implementation details

### D.1   Architecture of GNNs

We describe the details of the GNN architectures used in all numerical experiments. The first convolutional layer takes $H_0$-dimensional node embedding vectors, $\boldsymbol{h}_{\boldsymbol{\theta}}^0$ for each node, as input, yielding $H_1$-dimensional feature vectors $\boldsymbol{h}_{\boldsymbol{\theta}}^1$. Then, the ReLU function is applied as a component-wise nonlinear transformation. The second convolutional layer takes the $H_1$-dimensional feature vectors, $\boldsymbol{h}_{\boldsymbol{\theta}}^1$, as input, producing a $H^{(2)}$-dimensional vector $\boldsymbol{h}_{\boldsymbol{\theta}}^2$. Finally, a sigmoid function is applied to the $H^{(2)}$-dimensional vector $\boldsymbol{h}_{\boldsymbol{\theta}}^2$, and the output is the soft solution $\boldsymbol{p}_{\boldsymbol{\theta}} \in [0, 1]^N$. As in [Schuetz et al., 2022a], we set $H_0 = \text{int}(N^{0.8})$ or , $H_1 = \text{int}(N^{0.8}/2)$ and $H^2 = 1$ for both GCN and GraphSAGE .

### D.2   Training settings

For all numerical experiments, we use the AdamW [Kingma and Ba, 2014] optimizer with a learning rate of $\eta = 10^{-4}$ and weight decay of $10^{-2}$, and we train the GNNs for up to $10^5$ epochs with early stopping set to the absolute tolerance $10^{-5}$ and patience $10^3$. As discussed in Schuetz et al. [2022a], the GNNs are initialized with five different random seeds for a single instance because the results are dependent on the initial values of the trainable parameters; thus selecting the best solution.

## E   Experiment details

### E.1   Benchmark problems

**MIS problems.**   There are some theoretical results for MIS problems on RRGs with the node degree set to $d$, where each node is connected to exactly $d$ other nodes. The MIS problem is a

fundamental NP-hard problem [Karp, 2010] defined as follows. Given an undirected graph $G(V, E)$, an independent set (IS) is a subset of nodes $\mathcal{I} \in V$ where any two nodes in the set are not adjacent. The MIS problem attempts to find the largest IS, which is denoted $\mathcal{I}^*$. In this study, $\rho$ denotes the IS density, where $\rho = |\mathcal{I}|/|V|$. To formulate the problem, a binary variable $x_i$ is assigned to each node $i \in V$. Then the MIS problem is formulated as follows:

$$f(\boldsymbol{x}; G, \lambda) = -\sum_{i \in V} x_i + \lambda \sum_{(i,j) \in E} x_i x_j,$$

where the first term attempts to maximize the number of nodes assigned 1, and the second term penalizes the adjacent nodes marked 1 according to the penalty parameter $\lambda$. In our numerical experiments, we set $\lambda = 2$, following Schuetz et al. [2022a], no violation is observed as in [Schuetz et al., 2022a]. First, for every $d$, a specific value $\rho_d^*$, which is dependent on only the degree $d$, exists such that the independent set density $|\mathcal{I}^*|/|V|$ converges to $\rho_d^*$ with a high probability as $N$ approaches infinity [Bayati et al., 2010]. Second, a statistical mechanical analysis provides the typical MIS density $\rho_d^{\text{Theory}}$, as shown in Fig. 2, and we clarify that for $d > 16$, the solution space of $\mathcal{I}$ undergoes a clustering transition, which is associated with hardness in sampling [Barbier et al., 2013] because the clustering is likely to create relevant barriers that affect any algorithm searching for the MIS $\mathcal{I}^*$. Finally, the hardness is supported by analytical results in a large $d$ limit, which indicates that, while the maximum independent set density is known to have density $\rho_{d \to \infty}^* = 2 \log(d)/d$, to the best of our knowledge, there is no known algorithm that can find an independent set density exceeding $\rho_{d \to \infty}^{\text{alg}} = \log(d)/d$ [Coja-Oghlan and Efthymiou, 2015].

**MaxCut problems.**   The MaxCut problem is also a fundamental NP-hard problem [Karp, 2010] with practical application in machine scheduling [Alidaee et al., 1994], image recognition [Neven et al., 2008] and electronic circuit layout design [Deza and Laurent, 1994]. The MaxCut problem is also a fundamental NP-hard problem [Karp, 2010] Given an graph $G = (V, E)$, a cut set $\mathcal{C} \in E$ is defined as a subset of the edge set between the node sets dividing $(V_1, V_2 \mid V_1 \cup V_2 = V,\ V_1 \cap V_2 = \emptyset)$. The MaxCut problems aim to find the maximum cut set, denoted $\mathcal{C}^*$. Here, the cut ratio is defined as $\nu = |\mathcal{C}|/|\mathcal{V}|$, where $|\mathcal{C}|$ is the cardinality of the cut set. To formulate this problem, each node is assigned a binary variable, where $x_i = 1$ indicates that node $i$ belongs to $V_1$, and $x_i = 0$ indicates that the node belongs to $V_2$. Here, $x_i + x_j - 2x_i x_j = 1$ holds if the edge $(i, j) \in \mathcal{C}$. As a result, we obtain the following:

$$f(\boldsymbol{x}; G) = \sum_{i<j} A_{ij}(2x_i x_j - x_i - x_j).$$

This study has also focused on the MaxCut problems on $d$-RRGs, for which several theoretical results have been established. Specifically, for each $d$, the maximum cut ratio is given by $\nu_d^* \approx d/4 + P_* \sqrt{d/4} + \mathcal{O}(\sqrt{d})$, where $P_* = 0.7632 \dots$ with a high probability as $N$ approaches infinity [Parisi, 1980, Dembo et al., 2017]. Thus, we take $\nu_d^{\text{UB}} = d/4 + P_* \sqrt{d/4}$ as an upper bound for the maximum cut ratio in the large $n$ limit.

**DBM problems.**   Here, the topologies are taken from the Cora citation network [Sen et al., 2008], where each node has 1,433 bag-of-words features, and each edge represents likelihood, as predicted by a machine learning model. Mandi et al. [2022] focused on disjoint topologies within the given topology, and they created 27 distinct instances with varying properties. Each instance comprises 100 nodes representing scientific publications, divided into two groups of 50 nodes $N_1$ and $N_2$. The optimization task is to find the maximum matching, where diversity constraints ensure connections among papers in the same field and between papers of different fields. This is formulated using a penalty method as follows.

$$l(\boldsymbol{x}; C, M, \boldsymbol{\lambda}) = -\sum_{ij} C_{ij} x_{ij} + \lambda_1 \sum_i \text{ReLU}\left(\sum_j x_{ij} - 1\right) + \lambda_2 \sum_j \text{ReLU}\left(\sum_i x_{ij} - 1\right)$$
$$+ \lambda_3 \text{ReLU}\left(p \sum_{ij} x_{ij} - \sum_{ij} M_{ij} x_{ij}\right) + \lambda_4 \text{ReLU}\left(q \sum_{ij} x_{ij} - \sum_{ij} (1 - M_{ij}) x_{ij}\right),$$

where $C \in \mathbb{R}^{N_1 \times N_2}$ represents the likelihood of a link between each pair of nodes, an indicator $M_{ij}$ is set to 0 if article $i$ and $j$ share the same subject field (1 otherwise) $\forall i \in N_1$, and $j \in N_2$. The parameters $p, q \in [0, 1]$ represent the probability of pairs sharing their field and of unrelated pairs, respectively. As in Mandi et al. [2022], we explore two variations of this problem, with $p = q =$ being 25% and 5%, respectively, and these variations are referred to as Matching-1 and Matching-2, respectively. In this experiment, we set $\lambda_1 = \lambda_2 = 10$ and $\lambda_3 = \lambda_4 = 25$.

## E.2 GNNs

A GNN [Gilmer et al., 2017, Scarselli et al., 2008] is a specialized neural network for representation learning of graph-structured data. GNNs learn a vectorial representation of each node in two steps, i.e., the aggregate and combine steps. The aggregate step employs a permutation-invariant function to generate an aggregated node feature, and in the combine step, the aggregated node feature is passed through a trainable layer to generate a node embedding, known as "message passing" or the "readout phase." Formally, for a given graph $G = (V, E)$, where each node feature $\boldsymbol{h}_v^0 \in \mathbb{R}^{H_0}$ is attached to each node $v \in V$, the GNN updates the following two steps iteratively. First, the aggregate step at each $l$-th layer is defined as follows:

$$\boldsymbol{a}_v^l = \text{Aggregate}_{\boldsymbol{\theta}}^l \left( \{ h_u^{l-1}, \forall u \in \mathcal{N}_v \} \right),$$

where the neighborhood of $v \in V$ is denoted $\mathcal{N}_v = \{ u \in V \mid (v, u) \in E \}$, $\boldsymbol{h}_u^{l-1}$ is the node feature of the neighborhood, and $\boldsymbol{a}_v^l$ is the aggregated node feature of the neighborhood. Then, the combined step at each $l$-th layer is defined as follows:

$$\boldsymbol{h}_v^l = \text{Combine}_{\boldsymbol{\theta}}^l(\boldsymbol{h}_v^{l-1}, \boldsymbol{a}_v^l),$$

where $\boldsymbol{h}_v^l \in \mathbb{R}^{H_l}$ denotes the node representation at the $l$-th layer. Here, the hyperparameters for the total number of layers $L$ and the intermediate vector dimension $N^l$ are determined empirically. Although numerous implementations of GNN architectures have been proposed to date, the most basic and widely used architecture is the GCN [Scarselli et al., 2008], which is given as follows:

$$\boldsymbol{h}_v^l = \sigma \left( W^l \sum_{u \in \mathcal{N}(v)} \frac{\boldsymbol{h}_u^{l-1}}{|\mathcal{N}(v)|} + B^l \boldsymbol{h}_v^{l-1} \right),$$

where $W^l$ and $B^l$ are trainable parameters, $|\mathcal{N}(v)|$ serves as a normalization factor, and $\sigma : \mathbb{R}^{H_l} \to \mathbb{R}^{H_l}$ is some component-wise nonlinear activation function, e.g., the sigmoid or ReLU function.

# F  Additional experiments

## F.1  Numerical validation of practical issues presented in Section 3.1

In this sectioin, we will examine the issue (I) with continuous relaxations and the issue (II), the difficulties of optimization, as pointed out by previous studies [Wang and Li, 2023, Angelini and Ricci-Tersenghi, 2023], in the NP-hard problems of MIS and the MaxCut problem. Therefere, we conducted numerical experiments using the PI-GNN solver for MIS and MaxCut problems on RRGs with higher degrees. To ensure the experimental impartiality, we adhered to the original settings of the PI-GNN solver [Schuetz et al., 2022b]. Refer to Section E for the detailed experimental settings. Fig. 7 (top) shows the solutions obtained by the PI-GNN solver as a function of the degree $d$ for the MIS and MaxCut problems with varying system sizes $N$. These results indicate that finding independent and cut sets becomes unfeasible as the RRG becomes denser. In addition, to clarify the reasons for these failures, we analyzed the dynamics of the cost function for MIS problems with $N = 10,000$, with a specific focus on a graph with degrees $d = 5$ and $d = 20$, as depicted in Fig. 7 (bottom). For the $d = 5$ case, the cost function goes over the plateau of $\hat{l}(\boldsymbol{\theta}; G, \boldsymbol{\lambda}) = 0$ with $\boldsymbol{p}_\theta(G) = \boldsymbol{0}_N$, as investigated in the histogram, eventually yielding a solution comparable to those presented by Schuetz et al. [2022a]. Conversely, in the $d = 20$ case, the cost function remains stagnant on the plateau of $\hat{l}(\boldsymbol{\theta}; G, \boldsymbol{\lambda}) = 0$ with $\boldsymbol{p}_\theta(G) = \boldsymbol{0}_N$, thereby failing to find any independent nodes. Interpreting this phenomenon, we hypothesize that the representation capacity of the GNN is sufficiently large, leading us to consider the optimization of $\hat{L}_{\text{MIS}}(\boldsymbol{\theta}; G, \lambda)$ and $\hat{L}_{\text{MaxCut}}(\boldsymbol{\theta}; G)$ as a variational optimization problem relative to $\boldsymbol{p}_\theta$. In this case, $\boldsymbol{p}_\theta^* = \boldsymbol{0}_N$ satisfies the first-order variational optimality conditions $\delta \hat{l}_{\text{MIS}}/\delta \boldsymbol{p}_\theta|_{\boldsymbol{p}_\theta = \boldsymbol{p}^*} = \delta \hat{l}_{\text{MaxCut}}/\delta \boldsymbol{p}_\theta|_{\boldsymbol{p}_\theta = \boldsymbol{p}^*} = \boldsymbol{0}_N$, which implies a potential reason for absorption into the plateau. However, this does not reveal the conditions for the convergence to the fixed point $\boldsymbol{p}^*$ during the early learning stage or the condition to escape from the fixed point $\boldsymbol{p}^*$. Thus, an extensive theoretical evaluation through stability analysis remains an important topic for future work.

In summary, UL-based solver, minimizing $\boldsymbol{\theta}$ can be challenging and unstable. In particular, the PI-GNN solver, which is one of the UL-based solvers employing GNNs, fails to optimize $\boldsymbol{\theta}$ due to a

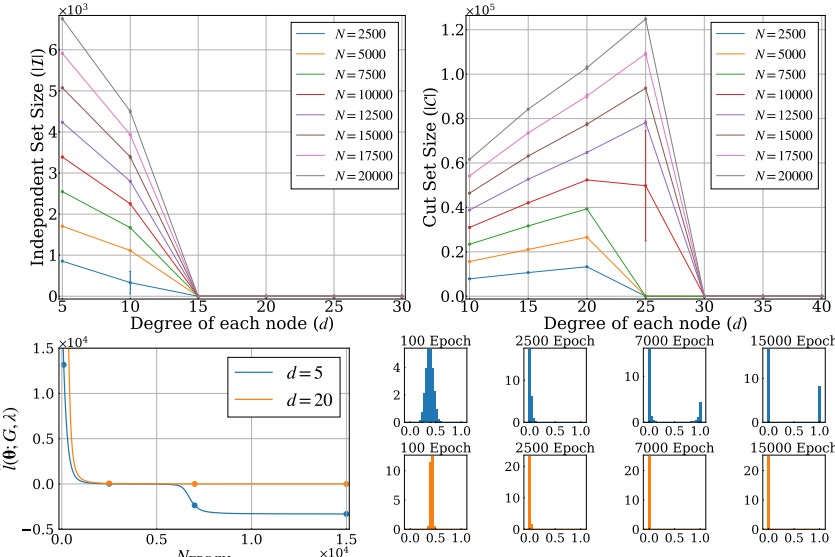

Figure 7: The top graph shows the independent set density for MIS problems (left) and the cut ratio for MaxCut problems (right) as a function of degree $d$ using the PI-GNN solver with varying system size $N$. Each data point represents the average result of five different graph instances, with the error bars indicating the standard deviation of those results. The bottom graph shows the cost as a function of the number of parameter updates $N_{\text{EPOCH}}$, for $N = 10000$ MIS problems on 5-RRG and 20-RRG. The histogram represents the relaxed vector distribution with varying numbers of parameter updates $N_{\text{EPOCH}}$. Each point in the bottom-left plot is linked to the corresponding bottom-right histogram.

local solution in complex CO problems on relatively dense graphs where the performance of greedy algorithms worsens. This issues can be potential bottleneck for more practical and relatively dense problems, making it challenging to employ the PI-GNN solver confidently.

## F.2 Additional results of MIS

We evaluate our method using the MIS benchmark dataset from recent studies [Goshvadi et al., 2023, Qiu et al., 2022], which includes graphs from SATLIB [Hoos and Stützle, 2000] and Erdős–Rényi graphs (ERGs) of varying sizes. Following Sun et al. [2023], our test set consists of 500 SATLIB graphs, each containing between 403 and 449 clauses with up to 1,347 nodes and 5,978 edges, 128 ERGs with 700 to 800 nodes each, and 16 ERGs with 9,000 to 11,000 nodes each. We conducted numerical experiments on PQQA using four different configurations: parallel runs with $S = 100$ or $S = 1000$ and shorter steps (3000 steps) or longer steps (30000 steps), similar to the approach in iSCO [Sun et al., 2023]. Table F.2 presents the solution quality and runtime results. The results show that CRA, which optimizes the relaxed variables as an optimization of GNN parameters, takes extra time for smaller ER-[700-800] instances due to the smaller number of decision variables. However, for larger instances, CRA achieves results comparable to iSCO. Although limited space makes it difficult to present other benchmark results employed by iSCO, such as MaxCut and MaxClique, numerical experiments on these benchmarks also show that CRA is less effective for small problems. However, for larger problems, the results are comparable to or slightly inferior to those of iSCO.

We also investigated the relationship between the order of the graph and the solving time of the solver, and the results are shown in Table F.2 and F.2. The results demonstrate that the runtime remains nearly constant as graph order and density increase, indicating effective scalability with denser graphs.

## F.3 Additional results of Gset

We conducted experiments across the additional GSET collection to further validate that including CRA enhances PI-GNN results beyond previously achievable in Table 6.

Table 3: ApR and runtime are evaluated on three benchmarks provided by DIMES [Qiu et al., 2022]. The ApR is assessed relative to the results obtained by KaMIS. Runtime is reported as the total clock time, denoted in seconds (s), minutes (m), or hours (h). The runtime and solution quality are sourced from iSCO [Sun et al., 2023]. The baselines include solvers from the Operations Research (OR) community, as well as data-driven approaches utilizing Reinforcement Learning (RL), Supervised Learning (SL) combined with Tree Search (TS), Greedy decoding (G), or sampling (S). Methods that fail to produce results within 10 times the time limit of DIMES are marked as N/A.

| Method | Type | ER-[700-800] | | ER-[9000-11000] | |
|---|---|---|---|---|---|
| | | ApR | Time | ApR | Time |
| KaMIS | OR | 1.000 | 52.13m | 1.000 | 7.6h |
| Gurobi | OR | 0.922 | 50.00m | N/A | N/A |
| Intel | SL+TS | 0.865 | 20.00m | N/A | N/A |
| | SL+G | 0.777 | 6.06m | 0.746 | 5.02m |
| DGL | SL+TS | 0.830 | 22.71m | N/A | N/A |
| LwD | RL+S | 0.918 | 6.33m | 0.907 | 7.56m |
| DIMES | RL+G | 0.852 | 6.12m | 0.841 | 5.21m |
| | RL+S | 0.937 | 12.01m | 0.873 | 12.51m |
| iSCO | fewer steps | 0.998 | 1.38m | 0.990 | 9.38m |
| | more steps | 1.006 | 5.56m | 1.008 | 1.25h |
| CRA | UL-based | 0.928 | 47.30m | 0.963 | 1.03h |

Table 4: ApR of the MIS problem on $\mathrm{RRG}(N, d)$. All the results are averaged based on 5 RRGs with different random seeds.

| Problem | ApR (CRA) | ApR (PI) | Time (CRA) | Time (PI) |
|---|---|---|---|---|
| $\mathrm{RRG}(1,000, 10)$ | 0.95 | 0.78 | 108 (s) | 98 (s) |
| $\mathrm{RRG}(1,000, 20)$ | 0.95 | 0.56 | 103 (s) | 92 (s) |
| $\mathrm{RRG}(1,000, 30)$ | 0.94 | 0.00 | 102 (s) | 88 (s) |
| $\mathrm{RRG}(1,000, 40)$ | 0.93 | 0.00 | 101 (s) | 82 (s) |
| $\mathrm{RRG}(1,000, 50)$ | 0.92 | 0.00 | 102 (s) | 82 (s) |
| $\mathrm{RRG}(1,000, 60)$ | 0.91 | 0.00 | 101 (s) | 91 (s) |
| $\mathrm{RRG}(1,000, 70)$ | 0.91 | 0.00 | 101 (s) | 86 (s) |
| $\mathrm{RRG}(1,000, 80)$ | 0.91 | 0.00 | 102 (s) | 93 (s) |
| $\mathrm{RRG}(5,000, 10)$ | 0.93 | 0.77 | 436 (s) | 287 (s) |
| $\mathrm{RRG}(5,000, 20)$ | 0.95 | 0.74 | 413 (s) | 280 (s) |
| $\mathrm{RRG}(5,000, 30)$ | 0.95 | 0.00 | 419 (s) | 283 (s) |
| $\mathrm{RRG}(5,000, 40)$ | 0.94 | 0.00 | 429 (s) | 293 (s) |
| $\mathrm{RRG}(5,000, 50)$ | 0.94 | 0.00 | 418 (s) | 324 (s) |
| $\mathrm{RRG}(5,000, 60)$ | 0.93 | 0.00 | 321 (s) | 302 (s) |
| $\mathrm{RRG}(5,000, 70)$ | 0.92 | 0.00 | 321 (s) | 325 (s) |
| $\mathrm{RRG}(5,000, 80)$ | 0.92 | 0.00 | 330 (s) | 305 (s) |

## F.4 Additional results of TSP

We conducted additional experiments on several TSP problems from the TSPLIB dataset[3], presenting results that illustrate the $\alpha$-dependency.

Experiments calculated the ApR as the ratio of the optimal value to the CRA result, with the ApR representing the average and standard deviation over 3 seeds. The "–" symbol in PI-GNN denotes cases where most variables are continuous values and where no solution satisfying the constraint was found within the maximum number of epochs. The same GNN and optimizer settings were used as in the main text experiments in Section 5.1.

---

[3]http://comopt.ifi.uni-heidelberg.de/software/TSPLIB95/.

Table 5: The ApR of the MIS problem on $\mathrm{ERG}(N, d)$ is evaluated against the results of KaMIS. Due to time limitations, the maximum running time for KaMIS was constrained. The results below present the average ApRs and runtimes across five different random seeds.

| Problem | CRA (ApR) | PI (ApR) | Time (CRA) | Time (PI) | Time (KaMIS) |
|---|---|---|---|---|---|
| $\mathrm{ERG}(1,000, 0.05)$ | 0.97 | 0.01 | 103 (s) | 98 (s) | 100 (s) |
| $\mathrm{ERG}(1,000, 0.10)$ | 0.95 | 0.00 | 100 (s) | 98 (s) | 210 (s) |
| $\mathrm{ERG}(1,000, 0.15)$ | 0.94 | 0.00 | 100 (s) | 92 (s) | 315 (s) |
| $\mathrm{ERG}(1,000, 0.20)$ | 0.91 | 0.00 | 99 (s) | 88 (s) | 557 (s) |
| $\mathrm{ERG}(1,000, 0.25)$ | 0.93 | 0.00 | 98 (s) | 82 (s) | 733 (s) |
| $\mathrm{ERG}(1,000, 0.30)$ | 0.90 | 0.00 | 98 (s) | 82 (s) | 1000 (s) |
| $\mathrm{ERG}(1,000, 0.35)$ | 0.92 | 0.00 | 99 (s) | 91 (s) | 1000 (s) |
| $\mathrm{ERG}(1,000, 0.40)$ | 0.91 | 0.00 | 97 (s) | 86 (s) | 1000 (s) |

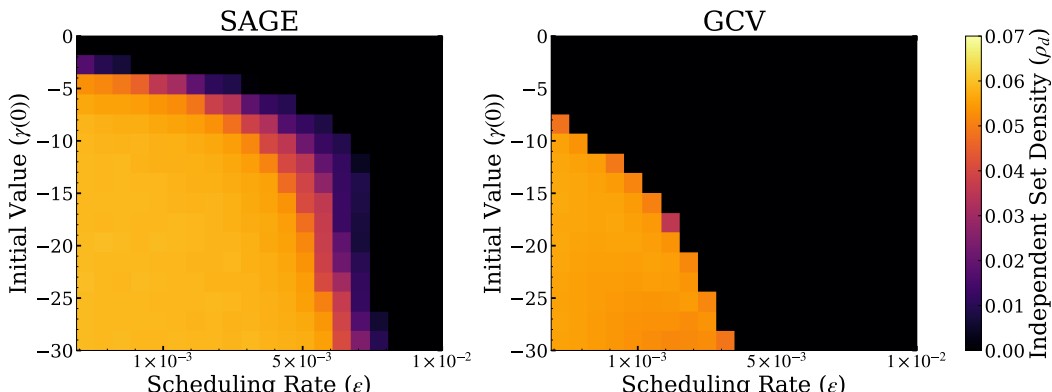

Figure 8: (Top) IS density of $N = 10,000$ MIS problems on 100-RRG as a function of initial scheduling $\gamma(0)$ and scheduling rate $\varepsilon$ values obtained by the CRA-PI-GNN solver using GraphSage (Left) and GCV (Right). The color of the heat map represents the average IS over five different instances.

Table F.4 shows that the CRA approach yielded solutions with an ApR exceeding 0.9 across various instances. Notably, for the burma14 problem, our method identified the global optimal solution (3,323) multiple times. However, the optimal value can vary based on the specific GNN architecture and problem structure, indicating that a more comprehensive ablation study could provide valuable insights in future work.

### F.5 Ablation over initial scheduling value and scheduling rate

We conducted an ablation study focusing on the initial scheduling value $\gamma(0)$ and scheduling rate $\varepsilon$. This numerical experiment was conducted under the configuration described in Section 5 and E. Fig. 8 shows the IS density of $N = 10000$ MIS problems on a 100-RRG as a function of the initial scheduling value $\gamma(0)$ and the scheduling rate $\varepsilon$ using the CRA-PI-GNN with both GraphSage and GCV. As can be seen, smaller initial scheduling $\gamma(0)$ and scheduling rate $\varepsilon$ values typically yield better solutions. However, the convergence time increases progressively as the initial scheduling $\gamma(0)$ and scheduling rate $\varepsilon$ values become smaller. In addition, GraphSage consistently produces better solutions even with relatively larger initial scheduling $\gamma(0)$ and scheduling rate $\varepsilon$ values, which implies that the GNN architecture influences both the solution quality and the effective regions of the initial scheduling $\gamma(0)$ and scheduling rate $\varepsilon$ values for the annealing process.

### F.6 Ablation over curve rate

Next, we investigated the effect of varying the curvature $\alpha$ in Eq. 3.2. Numerical experiments were performed on MIS problems with 10,000 nodes and the degrees of 5 and 20, as well as MaxCut problems with 10,000 nodes and the degrees of 5 and 35. The GraphSAGE architecture was employed,

Table 6: ApR for MaxCut on Gset

| GRAPH | (NODES, EDGES) | GREEDY | SDP | RUN-CSP | PI-GNN | CRA |
|---|---|---|---|---|---|---|
| G1 | (800, 19,176) | 0.942 | 0.970 | 0.979 | 0.978 | **1.000** |
| G2 | (800, 19,176) | 0.951 | 0.970 | 0.981 | 0.976 | **0.998** |
| G3 | (800, 19,176) | 0.945 | 0.972 | 0.982 | 0.972 | **1.000** |
| G4 | (800, 19,176) | 0.949 | 0.971 | 0.980 | 0.978 | **0.999** |
| G5 | (800, 19,176) | 0.949 | 0.970 | 0.980 | 0.978 | **1.000** |
| G14 | (800, 4,694) | 0.946 | 0.952 | 0.956 | 0.988 | **0.994** |
| G15 | (800, 4,661) | 0.939 | 0.958 | 0.952 | 0.980 | **0.992** |
| G16 | (800, 4,672) | 0.948 | 0.958 | 0.953 | 0.965 | **0.990** |
| G17 | (800, 4,667) | 0.946 | – | 0.956 | 0.967 | **0.990** |
| G22 | (2,000, 19,990) | 0.923 | – | 0.972 | 0.987 | **0.998** |
| G23 | (2,000, 19,990) | 0.927 | – | 0.973 | 0.968 | **0.997** |
| G24 | (2,000, 19,990) | 0.927 | – | 0.973 | 0.959 | **0.998** |
| G25 | (2,000, 19,990) | 0.929 | – | 0.974 | 0.974 | **0.998** |
| G26 | (2,000, 19,990) | 0.924 | – | 0.974 | 0.965 | **0.998** |
| G35 | (2,000, 11,778) | 0.942 | – | 0.953 | 0.968 | **0.990** |
| G36 | (2,000, 11,766) | 0.942 | – | 0.953 | 0.966 | **0.991** |
| G37 | (2,000, 11,785) | 0.946 | – | 0.950 | 0.966 | **0.997** |
| G38 | (2,000, 11,779) | 0.943 | – | 0.949 | 0.966 | **0.991** |
| G43 | (1,000, 9,990) | 0.928 | 0.968 | 0.976 | 0.966 | **0.995** |
| G44 | (1,000, 9,990) | 0.920 | 0.955 | 0.978 | 0.968 | **0.998** |
| G45 | (1,000, 9,990) | 0.930 | 0.950 | 0.979 | 0.961 | **0.998** |
| G46 | (1,000, 9,990) | 0.930 | 0.960 | 0.976 | 0.974 | **0.998** |
| G47 | (1,000, 9,990) | 0.931 | 0.956 | 0.976 | 0.972 | **0.997** |
| G48 | (3,000, 6,000) | **1.000** | **1.000** | **1.000** | 0.912 | **1.000** |
| G49 | (3,000, 6,000) | **1.000** | **1.000** | **1.000** | 0.986 | **1.000** |
| G50 | (3,000, 6,000) | **1.000** | **1.000** | 0.999 | 0.990 | **1.000** |
| G51 | (1,000, 5,909) | 0.949 | 0.960 | 0.954 | 0.964 | **0.991** |
| G52 | (1,000, 5,916) | 0.944 | 0.957 | 0.955 | 0.961 | **0.990** |
| G53 | (1,000, 5,914) | 0.945 | 0.956 | 0.950 | 0.966 | **0.992** |
| G54 | (1,000, 5,916) | 0.900 | 0.956 | 0.952 | 0.970 | **0.988** |
| G55 | (5,000, 12,498) | 0.892 | – | 0.978 | 0.983 | **0.991** |
| G58 | (5,000, 29,570) | 0.945 | – | 0.980 | 0.966 | **0.989** |
| G60 | (7,000, 17,148) | 0.889 | – | 0.980 | 0.945 | **0.991** |
| G63 | (7,000, 41,459) | 0.948 | – | 0.947 | 0.968 | **0.989** |
| G70 | (10,000, 9,999) | 0.886 | – | 0.970 | 0.982 | **0.992** |

Table 7: Comparison of ApR performance across different values of $p$ for various TSPLIB instances.

|  | burma14 | ulysses22 | st70 | gr96 |
|---|---|---|---|---|
| ApR ($p = 2$) | $0.91 \pm 0.08$ | $0.89 \pm 0.03$ | $0.96 \pm 0.01$ | $0.81 \pm 0.05$ |
| ApR ($p = 4$) | $0.98 \pm 0.10$ | $0.92 \pm 0.02$ | $0.85 \pm 0.03$ | $0.82 \pm 0.03$ |
| ApR ($p = 6$) | $0.97 \pm 0.14$ | $0.88 \pm 0.07$ | $0.88 \pm 0.04$ | $0.90 \pm 0.05$ |
| ApR ($p = 8$) | $0.99 \pm 0.06$ | $0.89 \pm 0.05$ | $0.80 \pm 0.02$ | $0.86 \pm 0.05$ |
| ApR (PI) | $0.736 \pm 1.21$ | – | – | – |
| Optimum | 3,323 | 7,013 | 675 | 5,5209 |

with other parameters set as Section 5 adn E. As shown in Fig. 9, we observed that $\alpha = 2$ consistently yielded the best scores across these problems.

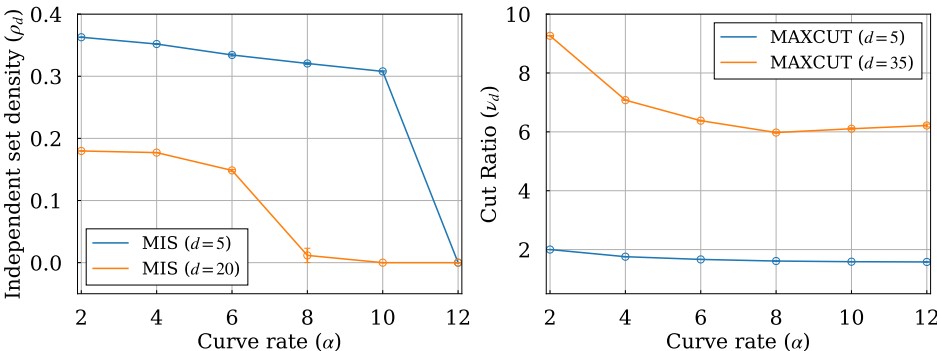

Figure 9: (Left) Independent set density as a function of curveture rate $\alpha$ in Eq. (3.2). (Right) Cut ratio as a function of curveture rate $\alpha$ in Eq. (3.2). Error bars represent the standard deviations of the results

