# OpenReview forum: "Controlling Continuous Relaxation for Combinatorial Optimization"
_NeurIPS.cc/2024/Conference — NeurIPS 2024 poster_

### Official Review · Reviewer_J6G3 · 2024-07-05

**Soundness:** 4
**Presentation:** 3
**Contribution:** 2
**Rating:** 6
**Confidence:** 4

**Summary:**

This article finds that the existing UL-solvers will trap into local optima and face rounding issues. This study proposes a continuous relaxation annealing (CRA) strategy and an auxiliary function to facilitate training.

**Strengths:**

1. The method proposed in the article is sound, easy to implement, and effective.
2. The article is well-written.

**Weaknesses:**

There are no major drawbacks in this article. There should be more reviews on neural combinatorial optimization solvers that apply annealing ideas as well (such as [1] provides annealing on the distance matrix of TSP).

[1] Lin, Xi, et al. ""Continuation path learning for homotopy optimization."" International Conference on Machine Learning. PMLR, 2023.

**Questions:**

1. Figure 8 provides parameter analysis for the N=10000 MIS problem. Can parameter sensitivity analysis be provided for other CO problems to demonstrate that CRA can be widely applied to general CO problems without special parameter design?
2. I am also concerned about the convergence speed under different parameter settings. Could you provide it as a function of the initial
scheduling and scheduling rate?
3. Can CRA be applied to routing problems such as TSP? If I understand correctly, the current $\phi$ function will have very small $p$-values in solving TSP, which may probably lead to a failure situation of CRA.

**Limitations:**

Well discussed.

---

> ### Author Rebuttal · Authors · 2024-08-07
>
> Thank you for your thorough and thoughtful review. We appreciate your recognition of our method's strengths, noting that it is "sound, easy to implement, and effective." Additionally, we are grateful for your comment that this article has no major drawbacks.
>
> **Weakness:**
> Thank you for your valuable suggestion regarding including a broader discussion on annealing methods. We acknowledge the importance of situating our work within the context of existing annealing-related research. The revised version will incorporate a comprehensive review of relevant annealing methods in the Related Work section, including the paper [1] you mentioned.
>
> **Question 1:**
> Thank you for your valuable question regarding parameter sensitivity analysis for other CO problems. For the MaxCut problem, we observed a similar sensitivity to parameters as in the MIS problem. Specifically, slower annealing rates result in better solution performance for Gset problems and MaxCut problems on RRGs, like simulated annealing. The revised version will include sensitivity analysis results for MaxCut and other non-d-regular random graphs in the Appendix. Ablation studies will accompany these results to demonstrate the robustness of CRA across various problem settings.
>
> **Question 2:**
> Thank you for your insightful question regarding the parameter sensitivity analysis. For the MIS problem, we confirmed that a linear change in $\gamma$ results in corresponding convergence speeds. To address your concern, we will include detailed results in the revised version showing the stopping time for different parameter settings and initial schedules. This will provide a clearer understanding of how the parameter settings affect the convergence speed across different problem instances.
>
> **Question 3:**
> Thank you for your insightful question regarding the applicability of CRA to the Traveling Salesman Problem (TSP). Our method can indeed be extended to this domain. However, the optimal $p$ value for solving TSP is unclear. In this study, our primary focus was on addressing the challenges of UL-based solvers highlighted by Wang et al. (2023) and achieving consistent results with PI-GNN. Consequently, we did not conduct experiments on TSP. Nonetheless, we recognize the importance of comprehensive comparisons for TSP and other problems. We plan to explore these areas in our future work, ensuring that our method's applicability and performance are thoroughly evaluated across a broader range of combinatorial optimization problems.
>
> We believe these revisions and the additional analyses will significantly enhance the robustness and comprehensiveness of our work. We kindly request you to consider this in your final evaluation.

---

> > ### Comment · Reviewer_J6G3 · 2024-08-10
> >
> > Thank you for your response.
> > ﻿
> > There are no major drawbacks to this article. I will increase the rating to 6 to support your acceptance.
> > ﻿
> > I still believe that exploring applications in routing problems such as TSP is important, and I hope to see this part in the rebuttal period and the following versions of your manuscript.

---

> ### Author Response · Authors · 2024-08-13
> **Reply to Reviewer J6G3**
>
> Thank you for recognizing the contribution of our paper, and we appreciate your decision to raise the score.
>
> Based on your insightful comments, we conducted additional experiments on several TSP problems from the TSPLIB dataset (http://comopt.ifi.uni-heidelberg.de/software/TSPLIB95/). Below, we present the results demonstrating the $p$-dependency of the solutions.
>
> | Instance   | ApR ($p=2$)          | ApR ($p=4$)          | ApR ($p=6$)          | ApR ($p=8$)          | ApR (PI)            | Optimum |
> |------------|----------------------|----------------------|----------------------|----------------------|---------------------|---------|
> | burma14    | 0.91 ± 0.08          | 0.98 ± 0.10          | 0.97 ± 0.14          | 0.99 ± 0.06          | 0.736 ± 1.21        | 3,323   |
> | ulysses22  | 0.89 ± 0.03          | 0.92 ± 0.02          | 0.88 ± 0.07          | 0.89 ± 0.05          | --                  | 7,013   |
> | st70       | 0.96 ± 0.01          | 0.85 ± 0.03          | 0.88 ± 0.04          | 0.80 ± 0.02          | --                  | 675     |
> | gr96       | 0.81 ± 0.05          | 0.82 ± 0.03          | 0.90 ± 0.05          | 0.86 ± 0.05          | --                  | 5,5209  |
>
>
> In these experiments, we calculated the ApR as the ratio of the optimal value to the CRA result, with the ApR representing the average and standard deviation over 3 seeds.
> The "--" in PI-GNN indicates that most variables are continuous values and that the maximum epoch did not obtain a solution satisfying the constraint.
> We used the same GNN and optimizer as in our experiments of the main text.
>
> The table shows that the CRA approach found solutions with an ApR exceeding 0.9 across various instances. Notably, for the "burma14" problem, our method found the global optimal solution, 3,323, several times. However, the optimal $p$ value may vary depending on the specific GNN architecture and problem structure, which suggests that a more comprehensive ablation study would be beneficial in future work.
>
> We will include these results and additional problem instances in the appendix or main text of the revised version.
> We hope that these enhancements, which demonstrate the broader applicability and robustness of our approach, will further highlight the contribution of our work. We would greatly appreciate it if these improvements could be taken into consideration during your final evaluation.

---

### Official Review · Reviewer_BuGW · 2024-07-08

**Soundness:** 2
**Presentation:** 2
**Contribution:** 3
**Rating:** 6
**Confidence:** 4

**Summary:**

The proposed approach is an optimization method for each graph over GNN parameters where each output corresponds to the likelihood of the node belonging to the solution. The objective function consists of a penalty term along with a parameter scheduled to control the non-convexity of the objective.

**Strengths:**

1- The convex annealing approach proposed in training that controls the level of non-convexity. This is a valid approach to avoid getting trapped in local minima where the solution sizes are not large.

2- Theoretical results of the limiting points of the proposed objective with different \gamma.

3- The "no-data" requirement makes this method mostly generalizable, depending on tuning a set of hyper-parameters for each graph distribution.

**Weaknesses:**

[Major Comments]

1- The need to solve graph-based NP-hard problems that are originally formulated as ILPs stems from the unscalability of these solvers. For example, the scalability of the MIS problem depends on the number of nodes and the number of edges in the graph. This needs to be the motivation instead of the issues encountered in UL-based solvers.

2- While the proposed approach does not require training data (labeled or unlabeled), there are several hyper-parameters. Tuning these hyper parameters is a challenge. Further discussion is needed here.

3- Getting trapped in local minima is not only the case in GNNs or PI-GNNs. It exists for any continuous relaxation of Problem 1. This is due to the non-convexity inherited in these formulations. For example, if we re-write Problem (3) in matrix form, we can see that the objective has a constant hessian equal to the adjacency matrix of the graph. If the magnitudes and signs of the eigen values vary significantly, then this indicates possible positive and negative curvatures in the loss landscape. Replacing x in Problem (3) with the output of a GNN does not guarantee changes. Although it may be possible that it will make some local minima avoidable by adaptive optimizers (such as ADAM), there is a possibility that this type of overparameterization would create unwanted local minima that do not result in any feasible solutions. Theoretically analyzing this is very complicated due to the use of a GNN. However, empirical investigation can be used to better motivate and understand the proposed approach.

4- Similar to the previous point, rounding issues existed even before GNNs. See the SDP relaxations of MIS [1] and MaxCut [2] and how their dependence on rounding techniques (e.g. spectral clustering [3]) often fails to obtain optimal solutions. Rewriting is needed here.

5- The Stationary point p* = 0_n was not discussed in Section 3.1. Furthermore, in line 208, it is 0_n, whereas in line 2016, it is 0_N.

6- How was the GW approximation applied for the MaxCut problem? This approximation requires a normalized random vector drawn from the standard Gaussian distribution. How many samples were drawn? Given the SDP solution, one can simply draw multiple samples and pick the best where the only requirement is matrix-vector multiplication. This runs extremely fast with (i) no parameters of a NW, and (ii) no hyper-parameters to tune. The scenarios where such approaches fail need to be the motivation to propose the over-parameterized approach with convex annealing.

7- Missing many "data-independent" baselines (methods that do not require pre-trained models (such as DIFUSCO [4]) or training data such as RL-based solvers (LwD [5])) for comparison such as ILP solvers (Gurobi, CPLEX, or CP-SAT [6]), sampling methods such as iSCO [7], SOTA heuristics such as ReduMIS [8], and differentiable solvers such as [9].

8- Why does the paper only consider d-regular graphs? How about the performance on other graphs? How does the run-time of this method scale in terms of the graph order and density? This is a major limitation of this work.

[Minor Comments]

1- What is script C in line 83?

2- What is I and J in the equation after line 86?

3- "nural" in line 106.

4- Cite equation 3. An example is [10]

5- Paragraph 149 to 151 is ill-sentenced.

6- “Indeed” in line 232.

7- Cite Potts variable optimization.

8- This study “employs” in line 237.

9- "are" in Appendix F.3 in line 247.

[References]

[1] On the shannon capacity of a graph. IEEE TIT, 1979.

[2] Improved approximation algorithms for maximum cut and satisfiability problems using semidefinite programming. JACM, 1995.

[3] A tutorial on spectral clustering. Springer, 2007.

[4] Difusco: Graph-based diffusion solvers for combinatorial optimization. NeurIPS, 2023.

[5] Learning What to Defer for Maximum Independent Sets. ICML, 2020.

[6] https://developers.google.com/optimization

[7] Revisiting sampling for combinatorial optimization. ICML, 2023.

[8] A differentiable approach to the maximum independent set problem using dataless neural networks. Neural Networks, 2022.

[9] A branch and bound algorithm for the maximum clique problem. Computers & operations research, 1992.

**Questions:**

See Weaknesses.

**Limitations:**

See Weaknesses.

---

> ### Author Rebuttal · Authors · 2024-08-07
>
> Thank you so much for reviewing our paper, appreciating the strengths in our approach, particularly noting the validity of the convex annealing method to control non-convexity and avoid local minima, as well as the theoretical results of the limiting points with different $\gamma$ and the generalizability due to the "no-data" requirement.
>
> **Strengths 1:**
> We are grateful for your recognition of the validity of our approach to avoiding local minima. As you commented, "This is a valid approach to avoid getting trapped in local minima where the solution sizes are not large." We appreciate your positive feedback. Our solver has demonstrated that its solution performance (ApR) does not deteriorate even as the problem size increases. In fact, as shown in Figure 4, the computational time increases sub-linearly, similar to PI-GNN, indicating good scalability compared to other solvers such as Gurobi.
> Furthermore, the ApR does not significantly deteriorate even with increasing problem size. CRA-PI-GNN has shown significant improvements for the largest MaxCut problem G70. Therefore, we believe that scalability for larger problem sizes is not an issue. Could you please clarify what size you consider large? We will present results and solution times for MIS and MaxCut problems of the sizes you expect during the discussion time or in the revised version. We will include further discussion in the revised version of the manuscript.
>
> **Weakness 1:**
> Thank you for pointing out that "the need to solve graph-based NP-hard problems that are originally formulated as ILPs stems from the unscalability of these solvers."
> We agree this is an important issue, even more so than the problems encountered in UL-based solvers. We will consider your comments in the revised manuscript.
>
> **Weakness 2:**
> Regarding the comment, "While the proposed approach does not require training data (labeled or unlabeled), there are several hyper-parameters. Tuning these hyper-parameters is a challenge. Further discussion is needed here." As you pointed out, our solver includes hyper-parameters related to the annealing start point and the annealing rate. As shown in Appendix Figure 8, better solutions can be obtained by starting the annealing process from a lower point and slowing down the annealing rate. This annealing mechanism is similar to those used in simulated annealing and its derived algorithms, which face similar issues. Therefore, we consider this problem common to our solver. Developing methods to determine these hyper-parameters adaptively is an important future work. Additionally, other GNN architectures and other hyper-parameter settings can potentially yield better results, but our default settings demonstrate that many problems can be solved with reasonable accuracy.
>
> **Weakness 3:**
> Thank you for your insightful comment, "if we re-write Problem (3) in matrix form, we can see that the objective has a constant hessian equal to the adjacency matrix of the graph. If the magnitudes and signs of the eigenvalues vary significantly, then this indicates possible positive and negative curvatures in the loss landscape." This observation is indeed correct. As you noted, non-convexity arises at the stage of continuous relaxation of the objective function in the main text (line 114) before introducing GNN. While formulating with GNN may introduce new undesirable local minima, we intuitively expect the objective function to be smoother by over-parameterizing the optimization problem of the relaxed variables with a higher-dimensional neural network. However, as you correctly pointed out, theoretically verifying this is very challenging due to the non-linearity of GNNs. Therefore, we use empirical studies to discuss the issues arising from parameterizing relaxed variables with GNNs and the plateau problem at $0_{N}$ in Appendix F1. Such empirical research has not been conducted by other studies using UL-based solvers such as PI-GNN or Wang (2022, 2023), and we consider this one of our contributions. While our empirical studies are not comprehensive, we acknowledge the importance of more extensive numerical experiments and examining the solution paths in high-dimensional spaces traversed by GNNs using techniques like PCA for dimension reduction. We consider this an important direction for future work.
>
> **Weakness 4:**
> In lines 49-55, we explain the rounding issues in general linear programming (Hoffman-Kruskal theorem). However, we agree that adding explanations regarding SDP relaxations for MIS and MaxCut would be beneficial. We will revise the introduction to include this context in the revised version.
>
> **Weakness 5:**
> Due to space constraints, detailed explanations of the stationary point $p^{\ast}=0_{N}$ are provided in Appendix F1, as mentioned in lines 186-187. In the revised version, we will ensure the main text includes sufficient details about the stationary point to be self-consistent. We are also grateful for pointing out the inconsistent use of $0_{N}$ and $0_{n}$.
>
> **Weakness 6:**
> The parameters for the Goemans-Williamson (GW) approximation are all set according to the settings in the PI-GNN paper's (approximate) polynomial-time GW algorithm.
> The implementation used the open-source CVXOPT solver with CVXPY as the modeling interface. In the revised version, we will provide detailed parameter settings in the Appendix to address your concerns.
>
> **Weakness 7:**
> Please refer to the "Numerical Experiments" section of the Unified Response for a comprehensive discussion.
>
> **Weakness 8:**
> Please refer to the "Runtime Scalability with Varying Graph Density" section of the Unified Response.
>
> Given this evidence, we have adequately addressed your concerns and kindly request you to reconsider your score based on these results.

---

> ### Comment · Reviewer_BuGW · 2024-08-11
> **Thank you for the efforts. Response to Authors**
>
> ### Response to Authors:
>
> - The experiments are evaluated on RRG and ER. The run-time of the proposed method scales only with the size of the graph, which I find impressive. Your response "runtime remains nearly constant as graph order and density increase" is not accurate as it is clear that the run-time is ~4X more when $n$ increased. In subsequent versions of the paper, I highly recommend including KaMIS and ILP solvers in these tables, as they are both known to scale with density, not just graph size. However, first, I find that the Apr in these results are low, and KaMIS should have been used to examine the solution quality. Second, in general, I still think that the proposed method is under-evaluated. The authors should consider using several random graph generators from NetWorkX, especially since it is a dataless method.
>
> - Discussing the intuition of over-parameterization to smoothen the landscape is acknowledged. I highly recommend that the authors include their response to this comment in the main body of the paper.
>
> - I acknowledge the additional experiments for MIS. The method clearly outperforms DIFUSCO while requiring no training data. However, the proposed approach still underperforms when compared to iSCO, a training-data-free sampling method where no GNNs are needed. However, I think that iSCO, while reporting significant results, is sensitive to the choice of the sampler and hyper-parameters. So overall, the CRA results in this table are fair.
>
> - **Question**: Why are the CRA MIS size results in the "Numerical Experiments" global response competitive, but the results in the attached PDF are not (way below 1)?
>
> - Larger graphs sizes (nodes) or harder instances can be any graph where ILPs and heuristics struggle in terms of either run-time or solution quality. For MIS, the authors can try mid-density (with respect to the complete graph) GNM graphs or ER (with $p > 0.4$) graphs.
>
> - Overall, I appreciate the dataless training approach and the theoretical support of the proposed method, and I encourage the authors to further investigate this direction. However, I feel that the paper could benefit from (i) a major revision in terms of writing (e.g., polishing the arguments about the non-convexity and rounding issues of general continuous relaxations of COPs) and (ii) a more comprehensive evaluation and explanation of the results. Therefore, I will only increase my score to 4.

---

> > ### Author Response · Authors · 2024-08-12
> > **Further response to reviewer BuGW**
> >
> > Thank you for raising your score, thoroughly reviewing it, and providing insightful comments.
> > We also thank you for your insightful understanding of over-parameterization.
> >
> > > However, first, I find that the Apr in these results are low, and KaMIS should have been used to examine the solution quality.
> >
> > We apologize for any confusion caused by the results in our global response's PDF.
> > **The results are based on IS density $\rho$ as defined in Line 99, not ApR.**
> > To clarify, we have included below a revised table where ApR is calculated, comparing our method against theoretical results [Barbier et al., 2023], as in Lines 299-302 in our experimental section.
> >
> > | Problem               | $\mathrm{ApR}$ (CRA) | $\mathrm{ApR}$ (PI) | Time (CRA) | Time (PI)  |
> > |-----------------------|--------------|-------------|------------|------------|
> > | $\mathrm{RRG}(1{,}000, 10)$ | 0.95         | 0.78        | 108 (s)    | 98 (s)     |
> > | $\mathrm{RRG}(1{,}000, 20)$ | 0.95         | 0.56        | 103 (s)    | 92 (s)     |
> > | $\mathrm{RRG}(1{,}000, 30)$ | 0.94         | 0.00        | 102 (s)    | 88 (s)     |
> > | $\mathrm{RRG}(1{,}000, 40)$ | 0.93         | 0.00        | 101 (s)    | 82 (s)     |
> > | $\mathrm{RRG}(1{,}000, 50)$ | 0.92         | 0.00        | 102 (s)    | 82 (s)     |
> > | $\mathrm{RRG}(1{,}000, 60)$ | 0.91         | 0.00        | 101 (s)    | 91 (s)     |
> > | $\mathrm{RRG}(1{,}000, 70)$ | 0.91         | 0.00        | 101 (s)    | 86 (s)     |
> > | $\mathrm{RRG}(1{,}000, 80)$ | 0.91         | 0.00        | 102 (s)    | 93 (s)     |
> > | $\mathrm{RRG}(5{,}000, 10)$ | 0.93         | 0.77        | 436 (s)    | 287 (s)    |
> > | $\mathrm{RRG}(5{,}000, 20)$ | 0.95         | 0.74        | 413 (s)    | 280 (s)    |
> > | $\mathrm{RRG}(5{,}000, 30)$ | 0.95         | 0.00        | 419 (s)    | 283 (s)    |
> > | $\mathrm{RRG}(5{,}000, 40)$ | 0.94         | 0.00        | 429 (s)    | 293 (s)    |
> > | $\mathrm{RRG}(5{,}000, 50)$ | 0.94         | 0.00        | 418 (s)    | 324 (s)    |
> > | $\mathrm{RRG}(5{,}000, 60)$ | 0.93         | 0.00        | 321 (s)    | 302 (s)    |
> > | $\mathrm{RRG}(5{,}000, 70)$ | 0.92         | 0.00        | 321 (s)    | 325 (s)    |
> > | $\mathrm{RRG}(5{,}000, 80)$ | 0.92         | 0.000       | 330 (s)    | 305 (s)    |
> >
> > As shown by these results, the ApR exceeds 0.9 for all values of $d$.
> >
> > Additionally, we have conducted further comparisons with KaMIS for the Erdos–Renyi graph, focusing on runtime and ApR, which is evaluated by comparing our method against KaMIS. Due to time limitations, we constrained the running time for KaMIS, and the results below show the average ApRs and runtimes across five different random seeds.
> >
> > | Problem                  | CRA($\mathrm{ApR}$) | PI($\mathrm{ApR}$) | Time (CRA) | Time (PI) | Time (KaMIS) |
> > |--------------------------|---------------------|--------------------|------------|-----------|--------------|
> > | $\mathrm{ERG}(1{,}000, 0.05)$ | 0.97                | 0.01               | 103 (s)    | 98 (s)    | 100 (s)      |
> > | $\mathrm{ERG}(1{,}000, 0.10)$ | 0.95                | 0.00               | 100 (s)    | 98 (s)    | 210 (s)      |
> > | $\mathrm{ERG}(1{,}000, 0.15)$ | 0.94                | 0.00               | 100 (s)    | 92 (s)    | 315 (s)      |
> > | $\mathrm{ERG}(1{,}000, 0.20)$ | 0.91                | 0.00               | 99 (s)     | 88 (s)    | 557 (s)      |
> > | $\mathrm{ERG}(1{,}000, 0.25)$ | 0.93                | 0.00               | 98 (s)     | 82 (s)    | 733 (s)      |
> > | $\mathrm{ERG}(1{,}000, 0.30)$ | 0.90                | 0.00               | 98 (s)     | 82 (s)    | 1000 (s)     |
> > | $\mathrm{ERG}(1{,}000, 0.35)$ | 0.92                | 0.00               | 99 (s)     | 91 (s)    | 1000 (s)     |
> > | $\mathrm{ERG}(1{,}000, 0.40)$ | 0.91                | 0.00               | 97 (s)     | 86 (s)    | 1000 (s)     |
> >
> > These results demonstrate that our method performs comparably to KaMIS. The revised manuscript will include a more thorough comparison of larger node cases in the main text or appendices.
> >
> > Given this evidence and the detailed comparisons, we believe we have addressed this significant concern. We respectfully request that you reconsider your score based on these results. We are committed to further enhancing our paper as suggested and will include a more detailed discussion and additional results in the revised version.

---

> > > ### Comment · Reviewer_BuGW · 2024-08-12
> > > **Response to Authors**
> > >
> > > I would like to thank the authors for their response to my question and addressing most of my concerns. The results of the attached PDF make more sense now. Initially, I was looking at the increasing $d$ and $p$ for density as these parameters indicate the density of the these graphs. I am happy to further increase my score.
> > >
> > > Final remarks:
> > >
> > > - While the run-time comparison with KaMIS in Table 2 is encouraging, I wouldn't call CRA solution quality comparable. KaMIS is a CPU-based C++ (or C) implementation. I wonder whether it can be designed to be faster with GPUs (just sharing a thought, not asking for more experiments). However, this method depends on MIS-specific graph reductions that would still be slower on denser graphs.
> > >
> > > - **Overall**, I feel that the authors should emphasize that their method is dataless (no training data is required) which, I believe, is a great advantage and should have been reflected in the title. I believe that over-parameterization and benefiting from the structure of a GNN are what makes the contribution valuable. To me, the benefit here is similar, in spirit, to the topic of lifting in optimization.

---

### Official Review · Reviewer_qMFs · 2024-07-15

**Soundness:** 3
**Presentation:** 3
**Contribution:** 2
**Rating:** 4
**Confidence:** 4

**Summary:**

This paper aims to tackle shortcomings of the existing unsupervised learning-based solvers for combinatorial optimization, namely the local optima issue and the rounding issue. It proposes a novel technique called continuous relaxation annealing (CRA) strategy which introduces an additional penalty term to smooth the non-convexity of the objective function. This strategy is empirically shown to not only enhance the solution quality but also accelerate the learning process.

**Strengths:**

1. This paper is an interesting study on the unsupervised-learning based approaches on CO problems. The proposed method is simple but proves to be quite effective.
2. The empirical evaluation shows that CRA achieves a consistent improvement over PI-GNN
3. The authors have conducted extensive qualitative and quantitative analysis to help understand the proposed method

**Weaknesses:**

1. My main concern lies in the technical contribution from this paper. The whole framework and empirical evaluation is built upon PI-GNN, which makes the observation and conclusion from this paper not generalizable.
2. I feel the research from this paper is kind of out-of-date. Check https://openreview.net/forum?id=ZMP0Bki9aK for SOTA results on the CO problems considered in this paper. In fact, [1] also mention that simulated annealing would perform better than GNN, but only greedy methods are used as baselines.


[1] Maria Chiara Angelini and Federico Ricci-Tersenghi. Modern graph neural networks do worse than classical greedy algorithms in solving combinatorial optimization problems like maximum independent set. Nature Machine Intelligence, 5(1):29–31, 2023.

**Questions:**

N/A

**Limitations:**

Yes.

---

> ### Author Rebuttal · Authors · 2024-08-07
>
> Thank you for reviewing our paper; we appreciate your recognition of the simplicity and effectiveness of our proposed method, the consistent improvement of CRA over PI-GNN, and the extensive qualitative and quantitative analysis.
>
> **Weakness (Contribution):**
> Please refer to "Main Contribution and Novelty" of the Unified Response for a detailed explanation of our contributions.
>
> **Weaknesss (Comparison with SOTA):**
> Please refer to the "Numerical Experiments" section of the Unified Response for a comprehensive discussion.
>
> **Regarding Paper [1]:**
> You referenced the paper [1] by Angelini and Ricci-Tersenghi, which reports results for regular random graph problems ($d=3$ and $d=5$) with $\mathrm{ApR}(d=3)=0.984$ and $\mathrm{ApR}(d=5)=0.981$ using SA. we indeed observed $\mathrm{ApR}(d=3) \approx 0.95$ and $\mathrm{ApR}(d=5)=0.93$. However, with CRA-PI-GNN, we achieved $\mathrm{ApR}(d=3)=0.983$ and $\mathrm{ApR}(d=5)=0.981$ for $N=10{,}000$ MIS problems, which are comparable to the SA's results reported by Angelini and Ricci-Tersenghi. However, with CRA-PI-GNN, we achieved $\mathrm{ApR}(d=3)=0.983$ and $\mathrm{ApR}(d=5)=0.981$ for $N=10{,}000$ MIS problems, which are comparable to the SA's results reported by Angelini and Ricci-Tersenghi.
>     Moreover, for $d=20$, SA's results are $\mathrm{ApR}(d=20)=0.968$, while our results show $\mathrm{ApR}(d=20)=0.963$, which is also comparable to SA's performance. Although a more detailed comparison with SA is essential, it is worth noting that SA relies on single bit-flip local transitions and cannot leverage GPU parallel computation, which exhibits notoriously poor efficiency in high dimensional spaces, e.g., $10^6 \sim 10^7$ variables.
>    In contrast, CRA-PI-GNN performs gradient-based optimization and thus efficiently utilizes GPUs and state-of-the-art optimizers, which can solve bigger problems with sub-linear computational costs, as shown in Figure 4 in the main text.
>
> Given the additional experiments we conducted and our significant contribution in addressing the limitations of data-independent UL-based solvers, as pointed out by the influential paper Wang (2023), we kindly request you to reconsider your score.

---

> ### Comment · Reviewer_qMFs · 2024-08-12
>
> I sincerely appreciate authors' for your response. However, I believe the link included in my second weakness part refers to another paper [1] that
> - achieves the SOTA performance
> - "performs gradient-based optimization and thus efficiently utilizes GPUs and state-of-the-art optimizers" exactly as you claimed for your method.
> - and also requires no supervision
>
> I do not see a clear discussion here.
>
> [1] Sun et al. Revisiting Sampling for Combinatorial Optimization. ICML 2023.

---

> ### Author Response · Authors · 2024-08-12
> **Reply to Reviewer qMFs**
>
> Thank you for your thoughtful feedback and continued engagement with our work.
>
> We apologize for any confusion caused by our previous explanation in the global (unified) response.
> The iSCO results presented in the *"Numerical Experiments"* of our global response are from the paper by Sun et al. (ICML 2023) [1].
> We apologize for any inconvenience and kindly request that you review these results in the global response, where we demonstrated CRA's performance that is comparable to or slightly below the SOTA iSCO method.
> As Reviewer BuGW noted, **"I acknowledge the additional experiments for MIS. The method clearly outperforms DIFUSCO while requiring no training data. However, the proposed approach still underperforms when compared to iSCO, a training-data-free sampling method where no GNNs are needed. However, I think that iSCO, while reporting significant results, is sensitive to the choice of the sampler and hyper-parameters. So overall, the CRA results in this table are fair."** This recognition of our comparison with iSCO supports the fairness of our results.
>
> It is also important to highlight that iSCO [1] uses a first-order Taylor expansion of the objective function to approximate the essential $\Delta(x)$ term in the discrete Langevin dynamics, enabling efficient parallel processing on GPUs.
> However, this method may experience performance degradation when the first-order approximation is inadequate. The iSCO paper primarily tests problems where the approximation is valid or nearly valid, which should be considered, particularly in black-box optimization with complex surrogate models.
>
> In contrast, our method does not rely on such approximations and can be applied as long as the gradient of the objective function, $\hat{l}(p; C, \lambda)$, is accessible. Indeed, Wang et al. (2022) [2] demonstrated the superiority of UL-based solvers in black-box optimization using complex surrogates with GNNs, leveraging the flexibility of UL-based solvers.
> Additionally, as Reviewer BuGW pointed out, **"I believe that over-parameterization and benefiting from the structure of a GNN are what makes the contribution valuable. To me, the benefit here is similar, in spirit, to the topic of lifting in optimization."**
> This suggests that solution performance could further improve depending on the GNN's over-parameterization and architecture.
> We believe that the statement in the *"Main Contribution and Novelty"* section of the Global Response, "Our primary contribution is the introduction of CRA to overcome the limitations of UL-based solvers (Type II), as highlighted by Wang et al. (ICLR 2023)," is crucial to realizing these future possibilities.
>
> To further strengthen our paper, we plan to include more detailed comparisons with iSCO in the revised version and additional discussions and results. We respectfully request that you reconsider your score based on these efforts.
>
> - [1] Sun et al. Revisiting Sampling for Combinatorial Optimization. ICML 2023.
> - [2] Wang et al. Unsupervised learning for combinatorial optimization with principled objective relaxation. NeurIPS 2022.

---

> ### Comment · Reviewer_qMFs · 2024-08-13
>
> Thank you for your additional response.
>
> Generally, I'm satisfied with the technical part of this work, but I'm largely confused by the takeaways from this paper as it is different from my intuition. There could be bias from my personal research, so I promise to authors that I will continue the discussion with other reviewers to see if I take it incorrectly.
>
> - Firstly, I'm confused why authors feel there is any essential difference between "a first-order Taylor expansion of the objective function" used in iSCO and the "gradient of the objective function" used in CRA. From my point of view, gradient is exactly an outcome of the first-order Taylor expansion. In iSCO, the parameter space spans the scores $p_i$ on all nodes/edges; and in CRA, the parameter space spans all the GNN parameters.
>
> - Then it leads to my second question. Clearly, iSCO owns more parameters than CRA when the graph size is even larger, then why does CRA outperform iSCO on large instances if the over-parameterization is its advantage against iSCO? My personal answer here is that iSCO is optimized on a larger parameter space and it converges slowly on large instances.
>
> The reason I stated that this research is out-of-date is because the method introduced by Wang (2023) is clearly worse than iSCO in both its performance and the interpretability. Given the recent advances in sampling-based method, either unsupervised (iSCO) or supervised (DFUSCO), I do question the significance of this work on the Type II solvers. I note that all the theoretical analyses are on the output parameters $p_i$, isn't this type of analyses more  related to gradient-based simulated annealing methods whose parameters are exactly $p_i$ here?
>
> I do agree with authors that "CRA can be easily generalized to UL-based solvers (Type I) and other relaxation-based solvers", which seems to be a more significant topic to me. But currently, I feel this work is quite restricted to a narrow problem where other methods have already stood out.

---

> ### Author Response · Authors · 2024-08-13
> **Reply to qMFs**
>
> Thank you very much for your continued engagement and for addressing our responses. We greatly appreciate your acknowledgment of CRA's generalizability and the significance of our work.
>
> > Firstly, I'm confused why authors feel there is any essential difference between "a first-order Taylor expansion of the objective function" used in iSCO and the "gradient of the objective function" used in CRA.
>
> We apologize for our unclear response. There might be a need for clarification regarding the iSCO [1]. We understand that **iSCO, which employs discrete Langevin Monte Carlo, is not a gradient descent algorithm based on continuous relaxation.**
> Instead, it generalizes Langevin dynamics (gradient flow) from continuous probability distributions to discrete ones.
> iSCO changes the temperature while numerically simulating the Equation from Theorem 3.1, characterized by Equation (23) on Sun 2022 [2].
>
> Executing discrete Langevin dynamics without approximation requires calculating the difference between the objective functions $f(y)$ and $f(x)$ for the next transition candidates, as mentioned in Section 3.2.1 of Sun 2023 [1]. This approach is noted to be computationally intensive, as Sun et al. themselves acknowledge in Section 3.2.1.
> **The need to compute the difference between objective values rather than gradients indicates that iSCO is not based on continuous relaxation**. Although they approximate this difference with a Taylor expansion to expedite the simulation, it remains unclear how effective this approximation is for more complex problems and how to related to our method.
>
> > Then it leads to my second question. Clearly, iSCO owns more parameters than CRA when the graph size is even larger, then why does CRA outperform iSCO on large instances if the over-parameterization is its advantage against iSCO? My personal answer here is that iSCO is optimized on a larger parameter space and it converges slowly on large instances.
>
> As explained above, **iSCO uses the gradient of the objective function to approximate and efficiently simulate original discrete Langevin dynamics without using continuous relaxation**.
> On the other hand, since CRA uses continuous relaxation, it is easy to implement gradient descent, and optimizers with good convergence properties, such as AdamW, can be used. Indeed, we used AdamW in our numerical experiments. Also, as reviewer BuGW pointed out, over-parameterizing the decision variables using GNNs may smooth the loss landscape, making it more suitable for optimization and speeding up the convergence speed.
> However, it is difficult to investigate this hypothesis at present theoretically, and we consider that a more detailed investigation is future work. Additionally, it is crucial to investigate iSCO's implementation details, including its specific code-level parallelization methods. Unfortunately, as the iSCO code has yet to be shared, it has been challenging to delve into these details during the discussion period.  In the revised version, we will add these differences between iSCO and CRA to the related work.
>
> > I note that all the theoretical analyses are on the output parameters $p_{i}$, isn't this type of analyses more related to gradient-based simulated annealing methods whose parameters are exactly $p_{i}$ here?
>
> > I do question the significance of this work on the Type II solvers.
>
> Our analysis is a theoretical analysis of the output parameters $p_{i}$.
> As you point out, there is an indirect relationship with gradient-based simulated annealing for $p_{i}$, but gradient-based simulated annealing is a method that performs Langevin Monte Carlo while lowering the temperature, and our method is a gradient descent method that controls the term $\Phi(p)$ that controls the continuity and discreteness.
> **Also, the method of simply performing continuous relaxation and gradient-based simulated annealing for $p_{i}$ cannot guarantee the original CO problems because they have the rounding issue, as pointed out in our main text.**
>
> We observed superior results compared to DIFUSCO for larger instances. As a heuristic approach, obtaining good solutions for problems too large for an exact solution method like Gurobi is vital. Also, as Reviewer BuGW pointed out, a more comprehensive comparison is needed between CRA and iSCO, including the setting of hyperparameters. We think that investigating which method is superior in a more quantitative way is future work (While we intended to conduct a detailed comparison with iSCO on larger problems such as d-regular random graphs during the rebuttal period, the unavailability of the iSCO code prevented us from completing this within the time limit).
>
> Considering these points, we would sincerely appreciate it if you could provide a final evaluation.
>
> - [1] Sun et al. Revisiting Sampling for Combinatorial Optimization. ICML 2023.
> - [2] Sun et al. Discrete langevin samplers via wasserstein gradient flow. AISTATS 2023.

---

### Official Review · Reviewer_fKtw · 2024-07-15

**Soundness:** 2
**Presentation:** 1
**Contribution:** 2
**Rating:** 4
**Confidence:** 1

**Summary:**

This paper presents a heuristic method for producing solutions to combinatorial optimization problems, which is based around solving a continuous relaxation of the problem. The main focus of the paper is on an additional penalty term to add to the objective of this relaxation which aims to reward solutions that are closer to satisfying the integrality constraints on the decision variables.

**Strengths:**

The computational study seems relatively comprehensive in that it studies a number of different problem settings in a fair amount of detail.

**Weaknesses:**

The paper is very dense and hard to follow, with little context provided to the reader. The method presented and evaluated in the computational study is ultimately an extension of the "PI-GNN" solver, but this fact is oddly kind of buried, with only an indirect reference in the introduction ("the solver that applies the CRA to the PI-GNN solver is referred to as the CRA-PI-GNN solver", with no indication that this is a main takeaway from the work), and then again at the end of Section 3.2. The paper does not explain in detail or formality what the PI-GNN solver is or how it works (how, specifically, does the CGA actually hook into PI-GNN?), and so a reader without prior familiarity cannot really understand or assess the new contributions laid out in Section 3. Ultimately, I do not feel confident that I can understand, and thus evaluate, the contributions proposed in the paper.

**Questions:**

The main contribution of the paper is an additional penalization term to induce solutions that are feasible w.r.t. the binary constraints on the decision variables, but I do not see explicit discussion in the Experiments section about the feasibility of the solutions produced (wr.t. both the integrality constraints and the other equality/inequality constraints). Are all of the solutions used in the computational study feasible for the "true" problem? If there are numerical tolerances used to "fudge" exact feasibility, what are these tolerance values?

---

> ### Author Rebuttal · Authors · 2024-08-07
>
> Thank you for your thorough and thoughtful review. We appreciate the time and effort you have invested in providing valuable feedback.
>
> **Understanding Difficult Sections:**
> Thank you for your insightful feedback and for pointing out the areas that need clarification.
> We understand the importance of addressing any difficulties you encountered while reading our paper. After introducing combinatorial optimization and continuous relaxation methods, Section 2.1 explains UL-based solvers, categorizing them into Type I (line 127) and Type II (line 138), explicitly stating that PI-GNN is Type II. Lines 149-151 explicitly state that this paper focuses on Type II UL-based solvers, specifically PI-GNN. Furthermore, Section 3.2 explains how CRA integrates into Type II UL-based solvers and why it is effective.
> Additionally, we have shared all the relevant codes for review to support our explanations further. Appendix E2 provides a detailed explanation of GNN, enhancing our understanding of its role and implementation within our framework. Please specify the exact parts that remain unclear. We would greatly appreciate detailed pointers to enhance our revisions and ensure our explanations are as clear and comprehensive as possible.
>
> **Question:**
> Thank you for pointing out, "I do not see explicit discussion in the Experiments section about the feasibility of the solutions produced."
> This comment is invaluable in improving the clarity of our presentation.
> The Experiments section needs to discuss the feasibility of the solutions and binary constraints more explicitly.
> As stated in lines 234-236,
> the annealing is performed until $\Phi(\theta; C) \approx 0$ for all numerical experiments.
> Accordingly, after the annealing, the values of the relaxed decision variables $p_{\theta}(C)$ became 0 or 1 within the 32-bit Floating Point range in Pytorch GPU.
> Additionally, as stated in lines 597-599, no violations of the constraints were observed in our numerical experiments.
> Thus, all results presented in Section 5 (Experiments) are feasible solutions.
> However, this have needed to be clarified in Section 5 (Experiments).
> Considering your comment, we will ensure that these clarifications are explicitly stated in the revised version's main text.
>
> Considering our responses, we kindly request you to reconsider your score. Additionally, please let us know if any specific parts remain difficult to understand.

---

### Author Rebuttal · Authors · 2024-08-07

## Unified response to all reviewers

We sincerely thank the reviewers for their thorough and insightful reviews. Reviewer J6G3 found our idea interesting and promising, and Reviewer fKtw appreciated the comprehensiveness of our numerical experiments. However, Reviewer qMFs and Reviewer BuGW expressed some confusion about our contributions and raised insightful questions. We apologize for any confusion caused and will do our best to address these issues in the revised manuscript.
We will first address the common question of two reviewers (Reviewer qMFs and Reviewer BuGW) in the unified response.

**Main Contribution and Novelty:**
Our primary contribution is the introduction of CRA to overcome the limitations of UL-based solvers (Type II), as highlighted by Wang et al. (ICLR 2023).
As mentioned in the introduction, Section 5.4 of (Wang et al. ICLR2023) highlights that UL-based solvers (Type II), which do not use training data or past history, face significant challenges due to local minima issues, making it difficult to obtain reasonable solutions without such data.
Our numerical results demonstrate that by using CRA and removing the rounding step, we can achieve better solutions than those obtained by meta-learning-based UL solvers that use training data and history. This result is expected to reignite interest and further development in data-independent UL-based solvers.
Moreover, while our study focuses on applying CRA to UL-based solvers (Type II), CRA can be easily generalized to UL-based solvers (Type I) and other relaxation-based solvers, potentially addressing the rounding issue in these methods. Theorem 3.1 can also be easily generalized for UL-based solvers (Type I).
We plan to explore these general applications of CRA in future work.

**Numerical Experiments:**
We first respectfully disagree with Reviewer qMFs' statement that "the research from this paper is kind of out-of-date."
Different from sampling-based methods like simulated annealing, UL-based solvers have significant potential for further improvement through black-box optimization (Wang et al., 2022) and advancements in optimizers and GNNs. While it is important to compare our work with the SOTA method (iSCO) and other data-independent solvers with different learning methods, our main contribution is to break through the limitations of independent UL-based solvers (Type II), as pointed out by Wang et al. (2023). Our numerical experiments demonstrate that CRA approach outperforms UL-based solvers (Type I) using meta-learning with training data and history proposed by Wang et al. (2023) and PI-GNN, the typical method of UL-based solvers (Type II). We believe that the benchmark problems and the baseline are sufficient to confirm our findings. However, recognizing the importance of thorough comparisons, we conducted additional numerical experiments using the default parameters from our main text. We compared our solver with the SOTA method (iSCO) and other data-independent solvers with different learning methods.
 Although the test was conducted in different execution environments, the average ApR for each instance is shown below for your reference, where we benchmarked CRA against a total of 144 instances and compared the ApR and runtime with SOTA (iSCO) and other solvers. Following iSCO's evaluation, we used the results from KaMIS to calculate ApR. Each value represents the average across multiple instances.

| Method  | Type | GPU  | ER-[700-800]      | ER-[9000-11000]  |
|---------|------|------|-------------------|------------------|
| KaMIS   | OR   | --   | 1.000 (52.13m)    | 1.000 (7.6h)     |
| Gurobi  | OR   | --   | 0.922 (50.00m)    | --               |
| Intel   | SL+TS| A100 | 0.865 (20.00m)    | --               |
|         | SL+G | A100 | 0.777 (6.06m)     | 0.746 (5.02m)    |
| DGL     | SL+TS| A100 | 0.830 (22.71m)    | --               |
| LwD     | RL+S | A100 | 0.918 (6.33m)     | 0.907 (7.56m)    |
| DIMES   | RL+G | A100 | 0.852 (6.12m)     | 0.841 (5.21m)    |
|         | RL+S | A100 | 0.937 (12.01m)    | 0.873 (12.51m)   |
| DIFUSCO | Diffusion | V100 | 0.916 (26.67m) | --             |
| iSCO    | fewer steps | A100 | **0.998** (5.85m) | 0.990 (9.38m) |
| iSCO    | more steps  | A100 | **1.001** (1.28m) | **1.008** (1.25h) |
| **CRA**     | UL-based | V100 | 0.928 (47.30m) | 0.963 (1.03h)   |

The results show that CRA, which optimizes the relaxed variables as an optimization of GNN parameters, takes extra time for smaller ER-[700-800] instances due to the smaller number of decision variables. However, for larger instances, CRA achieves results comparable to iSCO. Although limited space makes it difficult to present other benchmark results employed by iSCO, such as MaxCut and MaxClique, numerical experiments on these benchmarks also show that CRA is less effective for small problems. However, for larger problems, the results are comparable to or slightly inferior to those of iSCO. These results will be added to our revised version's main text or appendix.
Note that our solver and sampling-based solvers involve numerous hyperparameters, making it challenging to claim superiority definitively. While we understand the importance of a comprehensive comparison of several other solvers in various hyperparameter settings, we kindly request you reconsider the scores in light of our significant contributions to the advancement of UL-based solvers in this study.

**Runtime Scalability with Varying Graph Density**
We are grateful of Reviewer BuGW's insight full comment, "How about the performance on other graphs? How does the run-time of this method scale in terms of the graph order and density? This is a major limitation of this work."
As shown in the attached PDF, our experiments demonstrate that the runtime remains nearly constant as graph order and density increase, indicating effective scalability with denser graphs. We will include these findings in the revised manuscript.

---

### Decision · Program_Chairs · 2024-09-25

**Decision:**

Accept (poster)

**Comment:**

While the original submission exhibited some minor flaws (such as writing issues, lack of comparison on a routing problem like the TSP, more discussion in the experiments section),  the authors have already dealt with one (TSP) and committed to fixing the others in a camera ready version. I expect them to follow through on this commitment.

This paper presents a nice, simple idea, that works very well. I further emphasize that the writing quality is not good enough, but a quick check with a professional proofreader or LLM ought to fix that easily. Reviewer BuGW also provides a number of small corrections. The experiments, especially with the addition of the TSP in the review phase, provide a convincing argument that the simple annealing mechanism introduced is effective at solving a simple, but diverse set of combinatorial optimization problems. The unsupervised learning technique is a fascinating way forward that stands in contrast to many of the other learning-heavy proposals for neural combinatorial optimization.